

# Anomalous Luttinger equivalence between temperature and curved spacetime: From black holes to thermal quenches

Baptiste Bermond[1][⋆], Maxim N. Chernodub[2][†],
Adolfo G. Grushin[3][‡] and David Carpentier[1][∘]

**1** ENSL, CNRS, Laboratoire de Physique, F-69342 Lyon, France
**2** Institut Denis Poisson UMR 7013, Université de Tours, Tours, 37200, France
**3** Univ. Grenoble Alpes, CNRS, Grenoble INP, Institut Néel, 38000 Grenoble, France

⋆ baptiste.bermond@ens-lyon.fr , † maxim.chernodub@idpoisson.fr ,
‡ adolfo.grushin@neel.cnrs.fr , ∘ david.carpentier@ens-lyon.fr

## Abstract

Building on the idea of Tolman and Ehrenfest that heat has weight, Luttinger established a deep connection between gravitational fields and thermal transport. However, this relation does not include anomalous quantum fluctuations that become paramount in strongly curved spacetime. In this work, we revisit the celebrated Tolman-Ehrenfest and Luttinger relations and show how to incorporate the quantum energy scales associated with these fluctuations, captured by gravitational anomalies of quantum field theories. We point out that such anomalous fluctuations naturally occur in the quantum atmosphere of a black hole. Our results reveal that analogous fluctuations are also observable in thermal conductors in flat-space time provided local temperature varies strongly. As a consequence, we establish that the gravitational anomalies manifest themselves naturally in non-linear thermal response of a quantum wire. In addition, we propose a systematic way to identify thermal analogues of black hole's anomalous quantum fluctuations associated to gravitational anomalies. We identify their signatures in propagating energy waves following a thermal quench, as well as in the energy density of heating Floquet states induced by repeated thermal quenches.

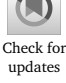

# 1   Introduction

Luttinger realized that if a gravitational field did not exist in nature, one could have invented it for the purposes of calculating thermal responses [1]. This idea can be traced back to the work by Tolman and Ehrenfest in the advent of general relativity, who noticed that in a time-independent curved spacetime, the temperature of black-body radiation is not spatially uni-

form even in thermal equilibrium [2,3]. Such a space-dependent temperature profile is known as the Tolman-Ehrenfest temperature. The insight of Luttinger was to suggest that thermal transport, thought of as a linear response of matter to a thermal gradient, can be derived by considering a counter-balancing weak gravitational field to restore equilibrium [1]. The Luttinger relation follows from the Tolman-Ehrenfest temperature, and is the groundbreaking idea that established the gravitational field as a key concept in the study of heat transport in materials [4–6].

A central idea behind Luttinger's and Tolman-Ehrenfest's relations is that heat has weight. Therefore heat contributes to the energy density, adding up to the energy density due to the rest-mass of a massive particle [7]. The consequences are particularly remarkable for relativistic massless particles which lack any intrinsic energy density scale. The only relevant energy density scale is set by the temperature, whose variations follows from those of spacetime.

However, in a strongly curved spacetime new energy density scales, that are absent in Luttinger's and Tolman-Ehrenfest's relations, appear. They manifest the anomalous thermodynamic behavior of quantum fluctuations induced by spacetime curvature. This interplay between geometry and vacuum fluctuations is analogous to Casimir effect, induced by a geometrical confinement instead of curvature [8, 9]. The fluctuations break the symmetries of the classical equation of motions, a phenomenon known as the gravitational anomalies. The appearance of these anomalous scales of energy density challenges us to understand their role in the equivalence relations between temperature and gravitational gradients, and their observable consequences for energy transport. Moreover, if these relations are modified, it is not evident what are their consequences beyond black-hole physics, for example in condensed matter. These are the questions that we address in this work.

In this work, we quantitatively evaluate the quantum corrections to the Tolman-Ehrenfest temperature, which in turn modify the celebrated Luttinger relation. We exemplify the importance of these corrections by focusing on 1+1 dimensional systems. Therefore, our results apply to the effective dynamics in reduced 1+1 dimension of quantum wires, but also of rotationally invariant systems such as isotropic black-holes, edge states of 2+1 dimensional topological gapped states of matter, or higher-dimensional systems in a strong magnetic field (see Fig. 1). We show that the Luttinger equivalence has to be corrected either when the curvature of spacetime is significant or, equivalently, when the local spatial variation of temperature is sizable. This equivalence allows us to observe that in several condensed matter systems the role of gravitational anomalies has been previously overlooked, and manifests through observable consequences. Consequently, we show that the seemingly elusive anomalous quantum fluctuations associated to the gravitational anomalies are detectable in experiments in flat-spacetime, beyond Weyl semimetals [10,11] or the thermal Hall effect [12]. In the outset we discuss that there is no fundamental obstruction to generalize our results to higher dimensional systems that possess gravitational anomalies.

At the technical level, in a relativistic massless theory in $1+1$ dimensions there is a single classical scale of energy density, set by the temperature $T$. As a consequence the energy density, $\varepsilon$, and pressure, $p$, are classically equal to each other. In other words the trace of the momentum-energy tensor, which quantifies the variation of energy upon a change of distance in space or time, vanishes. Incorporating the effects of anomalous fluctuations induced by spacetime curvature is achieved through the scale and Einstein anomalies. They generate two new scales of energy density $\varepsilon_q^{(1)}$ and $\varepsilon_q^{(2)}$. The first scale, $\varepsilon_q^{(1)}$, enters the non-vanishing trace of the momentum-energy tensor, $\varepsilon - p \propto \varepsilon_q^{(1)}$, a phenomenon known as the scale anomaly [13–16].

For massless particles, the energy density is set by the black-body radiation given by the Stefan-Boltzmann law $\varepsilon = p \propto \gamma T^2$ [17,18]. The second energy density scale introduced by the scale anomaly, $\varepsilon_q^{(2)}$, is an additive correction to $\varepsilon + p$. As a consequence, the appearance of

this energy density scale leads to a redefinition of the Tolman-Ehrenfest equilibrium temperature entering the Stefan-Boltzmann law. Remarkably, the exact same modified temperature is deduced from the correction of the off-diagonal component of the momentum-energy tensor, the energy current, by the so-called Einstein anomaly [19–21]: each chiral component of the current carries an energy proportional to the corrected equilibrium temperature instead of the bare Tolman-Ehrenfest temperature. Hence the effects of quantum fluctuations, captured either by the scale or by the gravitational anomaly correction to the momentum energy tensor, conspire to redefine the local equilibrium temperature in a curved spacetime. As a consequence, the Luttinger equivalence between a given temperature profile and a gravitational field has to be modified.

While field theory anomalies have been known to constrain non-Fermi liquids [22], and to be at the origin of various transport properties in condensed matter [23, 24], their interplay with the Luttinger equivalence was largely unexplored until now. When do these corrections matter? Strong spacetime curvature occurs naturally in the neighborhood of a black-hole. This curvature changes the nature of vacuum fluctuations in its vicinity, which ultimately lead to a radiating current of energy (Fig. 1(a)). Within 1+1 dimensional field theory, the relation between this Hawking's radiation and the anomalous quantum fluctuations was described using either the scale anomaly [25] or the Einstein anomaly [26]. We recall how both anomalies allow to define consistently a corrected equilibrium temperature [27], which vanishes at the black hole's horizon, as well as an outgoing energy current induced by the quantum fluctuations and whose asymptotic values identify with Hawking's radiation. The atmosphere of the black hole [28] corresponds to the soup of strong quantum fluctuations in which anomalous quantum corrections are strong.

Given that vacuum thermal effects close to a black hole are beyond experimental detection, we then consider anomalous fluctuations in condensed matter analogous to those in the black hole atmosphere. We show that, as a consequence of the modified Luttinger relation, such anomalous fluctuations occur where spatial variations of the temperature are large. First, we consider the historical domain of application of Luttinger's relation: that of thermal response theory. We show that a Kubo formula perturbative in the gravitational potential and its derivative accurately captures the effect of scale and gravitational anomalies. When translated in terms of temperature gradients, these anomaly-generated corrections are found to affect nonlinear thermal conductivities.

Stronger effects of anomalous fluctuations are expected beyond the realm of perturbative response theory. Inducing locally large fluctuations of energy reminiscent of those of a black hole's atmosphere requires strong local temperature variations. We consider a thermal quench occurring at the contact between two regions of different temperatures (Fig 1(c)). We show that local energy density's oscillations, as well as propagating heat waves resulting from the quench, recently identified within the conformal field theory framework [29–32], are a manifestation of the thermodynamics related to the anomalous Tolman-Ehrenfest temperature. Finally, we focus on a periodic sequence of metric quenches applied to a relativistic fermions (Fig 1(d)). This procedure induces a Floquet state recently described within Floquet conformal field theory [33–39]. While the total energy of this state increases exponentially, it concentrates on a few points [35, 36] which effectively behave as black holes [36]: the rate of increase of their energy is strongly corrected by quantum anomaly corrections, and the energy density is negative in their vicinity as in a black hole atmosphere.

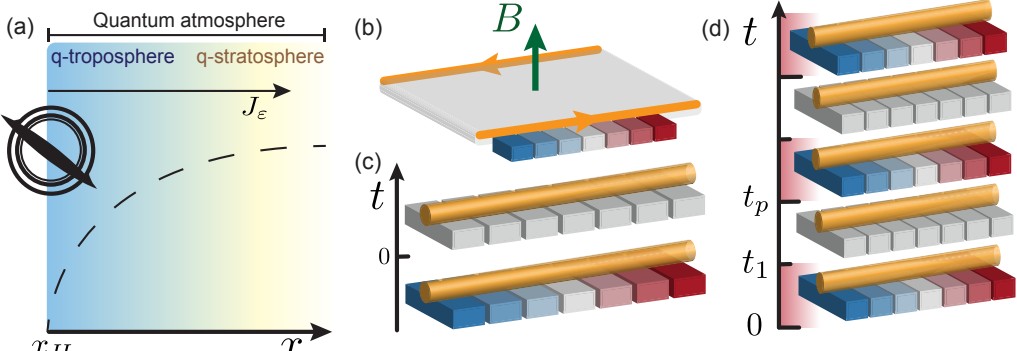

Figure 1: Four situations considered in this work where gravitational anomalies play a role through the anomalous Tolman-Ehrenfest temperature. (a) A black-hole's Hawking temperature at spatial infinity is fixed by an outgoing heat current $J_\varepsilon$ is determined by the gravitational anomalies: it originates from strong quantum fluctuations in a region close to the the horizon ($x_H$), a quantum atmosphere. (b) The edge of a 2D quantum Hall system hosts a 1+1 dimensional chiral edge mode. The difference between an externally imposed temperature profile, and the equilibrium temperature of the edge-mode is set by the anomalous Tolman-Ehrenfest temperature. (c) A quantum wire undergoes a quantum quench when an external temperature profile is suddenly switched off at time $t = 0$. The energy density profile following this quench displays oscillating features determined by gravitational anomalies. (d) A periodic dynamic is implemented by repeating over a period $t_p$ the previous thermal quench procedure. Within a heating phase, the energy density of the quantum wire increases exponentially, showing imprints of gravitational anomalies at spatially localized at points acting as black-hole analogues.

## 2 Anomalous Tolman-Ehrenfest temperature

### 2.1 Canonical Luttinger relation

We begin by recalling the derivation of Luttinger's relation [1,7] from Tolman-Ehrenfest's work [3]. Tolman and Ehrenfest realized that in a curved spacetime, the temperature of black-body radiation, and more generally of massless particles of velocity $v_F$, is not spatially uniform even in thermal equilibrium. In essence, thermal equilibrium in the presence of a gravitational field requires a non-uniform temperature profile to compensate for the red-shift experienced by radiation as it moves in the gravitational field. They showed that the equilibrium temperature can be inferred from a constant of motion with units of temperature, $T_0$, defined as

$$T(\boldsymbol{r})\sqrt{\xi^\mu(\boldsymbol{r})g_{\mu\nu}(\boldsymbol{r})\xi^\nu(\boldsymbol{r})} = T_0\,, \tag{1}$$

where $g_{\mu\nu}$ is the stationary background metric which depends on spatial coordinates $\boldsymbol{r}$, and $\xi^\mu$ is the time-like Killing vector. The constant $T_0$ is a reference temperature which has to be set, *e.g.* by boundary conditions. The Luttinger relation

$$\nabla_{\boldsymbol{r}}\phi = -\frac{\nabla_{\boldsymbol{r}}T}{T}\,, \tag{2}$$

is obtained from Eq. (1) by considering a time-independent metric

$$\mathrm{d}s^2 = e^{2\phi(\boldsymbol{r})}\,v_F^2\mathrm{d}t^2 - \mathrm{d}\boldsymbol{r}^2\,. \tag{3}$$

The metric is parametrized by a small dimensionless gravitational factor $\phi \equiv \Psi/c^2 \ll 1$, expressed, in a weak-gravity limit, in terms of a static gravitational potential $\Psi$ and the speed of light $c$.

By substituting Eq. (3) into Eq. (1), we obtain $T^2(r)g_{00}(r) = T_0^2$ which defines the Luttinger temperature (see Section 27 of [40]):

$$T^2(r)e^{2\phi(r)} = T_0^2 \,. \tag{4}$$

Upon spatial differentiating with respect to position we recover the Luttinger relation (2) This paves the way towards the correspondence within linear response framework between a perturbative parameter $\nabla\phi$ and the perturbative parameter $\nabla T/T$.

Neither Luttinger's nor Tolman-Ehrenfest's relations account for possible quantum anomalies. Our first goal is to show how these relations, widely used to identify the energy density and energy current of matter fields [4–6, 41–44], are modified in the presence of anomalies.

## 2.2 Tolman-Ehrenfest temperature of massless particles in $D = 1 + 1$ curved spacetime

For the sake of clarity, let us now restrict ourselves to a 1+1 dimensional space with coordinates $x^\mu = (x^0, x^1) \equiv (v_F t, x)$. We consider a general metric,

$$ds^2 \equiv g_{\mu\nu}dx^\mu dx^\nu = f_1(x)v_F^2 dt^2 - f_2(x)\,dx^2 \,, \tag{5}$$

defined in terms of two, time-independent, real and positive-valued functions $f_{1,2}(x)$ of the one-dimensional spatial coordinate. For convenience, we included the Fermi velocity $v_F$ of massless particles in the definition of the metric (5). In particular, the Luttinger metric (3) is recovered by considering the metric in the Fermi coordinate system:

$$f_1 \equiv g_{00} = e^{2\phi} \,, \qquad f_2 \equiv -g_{xx} = 1 \,, \tag{6}$$

which corresponds to a general relativistic generalization of an inertial coordinate frame [45]. A black-hole metric can also be captured by Eq. (5) by setting $f_1 = 1/f_2 = f$ with $f$ vanishing at the horizon.

We consider massless relativistic particles propagating with velocity $v_F$ in a curved 1+1 dimensional spacetime. The energy and momentum densities of these particles, as well as their associated current densities, are encoded in the energy-momentum tensor $\mathcal{T}_{\mu\nu}$. Spacetime translation invariance implies the conservation of this tensor at the classical level:

$$\nabla_\nu \mathcal{T}^\nu_\mu \equiv \frac{1}{\sqrt{-g}} \frac{\partial}{\partial x^\nu}\left(\mathcal{T}^\nu_\mu \sqrt{-g}\right) - \frac{1}{2}\frac{\partial g_{\alpha\beta}}{\partial x^\mu}\mathcal{T}^{\alpha\beta} - \frac{1}{2}\frac{\partial g_{\mu\alpha}}{\partial x^\nu}[\mathcal{T}^{\nu\alpha} - \mathcal{T}^{\alpha\nu}] = 0 \,. \tag{7a}$$

Moreover, the scale invariance of the theory implies that its trace vanishes:

$$\mathcal{T}^\mu_\mu = 0 \,. \tag{7b}$$

The diagonal components of this energy momentum tensor are the energy density $\varepsilon = \mathcal{T}^0_0$ and the pressure $p = \mathcal{T}^x_x$. Hence, scale invariance implies the equality $p = \varepsilon$. Finally, the Lorentz invariance implies the symmetry of the energy-momentum tensor:

$$\mathcal{T}^{\mu\nu} = \mathcal{T}^{\nu\mu} \,, \tag{7c}$$

manifesting that the density of the energy current $J_\varepsilon$ is proportional to the momentum density $\Pi$: $J_\varepsilon = v_F^2 \Pi$.

We consider a general stationary solution of equations (7):

$$[\mathcal{T}(x)]^\mu{}_\nu = \begin{pmatrix} \frac{C_0}{f_1} & \frac{C_1}{\sqrt{f_1 f_2}} \\ -\frac{f_2}{f_1}\frac{C_1}{\sqrt{f_1 f_2}} & -\frac{C_0}{f_1} \end{pmatrix}, \tag{8}$$

where $C_0, C_1$ are two constants. If we restrict ourselves to massless particles at equilibrium with a single local temperature, the only scale of density of energy or pressure is set by this equilibrium temperature through the extended Stefan-Boltzmann law [46–49]. The energy density $\varepsilon$ is obtained by summing the independent contributions $\varepsilon_\pm$ of left and right moving particles with respective central charges $c_\pm$

$$\varepsilon = \varepsilon_+ + \varepsilon_-, \quad \varepsilon_\pm = \frac{1}{2}c_\pm \gamma T^2, \quad \gamma = \frac{\pi k_B^2}{6\hbar v_F}. \tag{9}$$

Comparison with the diagonal terms of Eq. (8) leads to the relation

$$\frac{C_0}{f_1(x)} = \mathcal{T}^0{}_0 \equiv \varepsilon = -\mathcal{T}^x{}_x \equiv p. \tag{10}$$

The solution of Eq. (9,10) satisfies $T_{\text{TE}}^2(x)f_1(x) = 2C_0/[\gamma(c_+ + c_-)]$, which is independent of $x$. This turns out to be exactly the definition by Tolman and Ehrenfest of the equilibrium temperature (4):

$$T_{\text{TE}} = T_0\sqrt{f_1(x_0)/f_1(x)}, \tag{11}$$

where $T_0 = T(x_0)$ is an arbitrary reference temperature, commonly chosen where the metric is locally flat with $f_1(x_0) = 1$.

Alternatively, we can obtain this relation from the off-diagonal components of Eq. (8):

$$J_\varepsilon \equiv v_F\sqrt{f_1 f_2}\,\mathcal{T}^{x0} = v_F\sqrt{\frac{f_2}{f_1}}\mathcal{T}^x{}_0 = v_F\frac{C_1}{f_1}, \tag{12}$$

$$\Pi \equiv \frac{1}{v_F}\sqrt{f_1 f_2}\,\mathcal{T}^{0x} = -\frac{1}{v_F}\sqrt{\frac{f_1}{f_2}}\mathcal{T}^0{}_x = \frac{1}{v_F}\frac{C_1}{f_1}. \tag{13}$$

During their ballistic evolution, right and left moving particles don't exchange energies with each other. Each chiral species allows to define the local temperature through its local equilibrium chiral currents $J_{\varepsilon,\pm} = \pm v_F\,\varepsilon_\pm$. Combining this definition with Eq. (9) and Eq. (12) we again recover the Tolman-Ehrenfest relation (11). The net equilibrium current vanishes unless $c_+ \neq c_-$, for which

$$J_\varepsilon = v_F^2\Pi = (c_+ - c_-)\frac{\gamma v_F}{2}T^2 = (c_+ - c_-)\frac{\pi}{12\hbar}(k_B T)^2. \tag{14}$$

As a result, the equilibrium form of the classical momentum energy tensor (8) is expressed as follows:

$$[\mathcal{T}_{\text{cl}}(x)]^\mu{}_\nu = \begin{pmatrix} C_w & C_g\sqrt{\frac{f_1}{f_2}} \\ -C_g\sqrt{\frac{f_2}{f_1}} & -C_w \end{pmatrix} \times \frac{\gamma}{2}T_{\text{TE}}^2(x), \tag{15}$$

where we denoted

$$C_w = c_+ + c_-, \qquad C_g = c_+ - c_-. \tag{16}$$

### 2.3 Gravitational anomalies

Through (15), the previous section showed that both the equilibrium energy density, and the equilibrium chiral energy currents define the same classical Tolman-Ehrenfest temperature. Quantum mechanically, the density and currents are themselves constrained by the scale, translation and Lorentz invariances of the massless theory. All three symmetries are broken by quantum fluctuations of the matter field in a curved spacetime, a phenomenon known as a gravitational anomaly. Let us discuss the three gravitational anomalies affecting the momentum-energy tensor of massless matter in a curved spacetime.

**Trace anomaly.** The first is the scale anomaly which signals that the scale invariance of the single particle theory is broken by quantum fluctuations. The scale symmetry is a part of a larger, conformal symmetry group. Therefore, this anomaly is also called the conformal anomaly and, in different contexts, the Weyl or trace anomaly. As a consequence, Eq. (7b) no longer holds; the trace of the energy-momentum tensor of a scale-invariant classical theory does not vanish at the quantum level in a curved spacetime[1] [19, 59, 60]. The gravitational contribution to the trace anomaly is exact (it has no radiative corrections), and is determined by the Ricci scalar $R$ as follows:

$$\mathcal{T}^{\mu}_{\ \mu} = C_w \frac{\hbar v_F}{48\pi} R. \tag{17}$$

For the metric (5), $R$ simplifies to

$$R = \frac{\partial_x^2 f_1}{f_1 f_2} - \frac{1}{2} \frac{\partial_x f_1}{f_1 f_2} \left[ \frac{\partial_x f_1}{f_1} + \frac{\partial_x f_2}{f_2} \right]. \tag{18}$$

**Einstein anomaly.** Similarly, the spacetime translation invariance of the classical massless theory is broken at the level of the quantum field theory in a curved spacetime. It manifests the non-conservation of the energy current. In the case of a pure Einstein anomaly, *i.e.* while conserving Lorentz symmetry $\mathcal{T}^{\mu\nu} = \mathcal{T}^{\nu\mu}$, Eq. (7a) has to be replaced by

$$\nabla_\mu \mathcal{T}^{\mu\nu} = \frac{\hbar v_F}{96\pi} \frac{C_g}{\sqrt{|\det(g_{\rho\sigma})|}} \varepsilon^{\nu\mu} \nabla_\mu R, \tag{19}$$

with $\varepsilon^{0x} = 1$.

**Covariantly conserved tensor.** Finally, the Lorentz invariance of the classical theory, which results in the symmetry (7c) of the momentum-energy tensor, is also broken at the quantum level. However, the gravitational contribution of a pure Lorentz anomaly turns out to be equivalent to the graviational contribution of a pure Einstein anomaly, allowing to enforce either the Lorentz or the Einstein symmetry at the quantum level (see Sec. 12 of [19]). To show this equivalence, define a modified momentum-energy tensor $\tilde{\mathcal{T}}^{\mu\nu}$ from the $\mathcal{T}^{\mu\nu}$ resulting from the pure Einstein anomaly, given in of Eq. (19), according to

$$\tilde{\mathcal{T}}^{\mu\nu} = \mathcal{T}^{\mu\nu} + \frac{\hbar v_F}{96\pi} \frac{C_g}{\sqrt{|\det(g_{\rho\sigma})|}} \varepsilon^{\mu\nu} R. \tag{20}$$

---

[1]Notice that the trace anomaly can also appear in the flat spacetime in (self)interacting field theories as the consequence of energy dependence of the interaction couplings acquitted due to quantum fluctuation [50–52]. Contrary to the free theories in curved backgrounds that generate the exact trace anomaly (17), the interaction-induced trace anomaly involves the appropriate beta functions which are, in general, not one-loop exact [14]. We do not consider this manifestation of the trace anomaly while noticing that it may lead to various transport effects [53] including the Nernst-type thermal phenomena [54, 55] and non-topological boundary currents [56–58].

In this way we obtain a momentum-energy tensor satisfying the pure Lorentz anomaly, *i.e.* satisfying the energy-momentum conservation law $\nabla_\mu \tilde{\mathcal{T}}^{\mu\nu} = 0$, but with an anti-symmetric part that violates (7c):

$$\tilde{\mathcal{T}}^{\mu\nu} - \tilde{\mathcal{T}}^{\nu\mu} = \frac{\hbar v_F}{48\pi} \frac{C_g}{\sqrt{\left|\det\left(g_{\rho\sigma}\right)\right|}} \varepsilon^{\mu\nu} R. \tag{21}$$

These relations establish the equivalence between the Einstein and Lorentz anomalies.

## 2.4 Anomalous Tolman-Ehrenfest temperature

As shown in Sec. 2.2, the equilibrium temperature for massless matter in curved spacetime can be consistently defined (i) from thermodynamic quantities, the energy density and pressure, provided by the diagonal components of the momentum energy tensor, or (ii) from kinematic quantities, the density of energy current and momentum of left and right movers, given by the off-diagonal components of the momentum energy tensor. At the quantum level, diagonal and off-diagonal components get independently corrected by the scale anomaly on one side, and the Einstein - Lorentz anomalies on the other side. This immediately raises the question of whether a revised Tolman-Ehrenfest temperature can be defined incorporating the effects of quantum fluctuations. Quite remarkably, in this section we show that all three gravitational anomalies, while of different technical origins, concur to redefine the equilibrium temperature in a coherent manner, leading to an extended Luttinger equivalence.

**Anomalous momentum-energy tensor.** From the discussion in Sec. 2.3, we can choose without restriction to enforce the Lorentz symmetry at the quantum level, focusing on Einstein and Scale anomalies. Solving equations (17,19) for a symmetric tensor, we obtain

$$\mathcal{T} = \mathcal{T}_{\text{cl}} + \mathcal{T}_{\text{q}}, \tag{22}$$

where $\mathcal{T}_{\text{cl}}$ is the classical momentum energy tensor given by Eq. (15) and the quantum correction components are

$$[\mathcal{T}_{\text{q}}(x)]^\mu{}_\nu = \begin{pmatrix} \frac{C_w}{2}\left(\varepsilon_q^{(1)} + \varepsilon_q^{(2)}\right) & \frac{C_g}{2}\sqrt{\frac{f_1}{f_2}}\varepsilon_q^{(2)} \\ -\frac{C_g}{2}\sqrt{\frac{f_2}{f_1}}\varepsilon_q^{(2)} & \frac{C_w}{2}\left(\varepsilon_q^{(1)} - \varepsilon_q^{(2)}\right) \end{pmatrix}, \tag{23}$$

where $\varepsilon_q^{(1)}$ and $\varepsilon_q^{(2)}$ are the two new scales of energy density set by the quantum anomalies:

$$\varepsilon_q^{(1)} = \frac{\hbar v_F}{48\pi} R, \qquad \varepsilon_q^{(2)} = \frac{\hbar v_F}{48\pi}(R - 2\bar{R}), \tag{24}$$

with

$$2\bar{R} = \frac{1}{f_1(x)} \int_{x_0}^x \mathrm{d}y\, R(y)\partial_y f_1(y). \tag{25}$$

With the help of the expression (18) for the curvature $R$, we obtain

$$R - 2\bar{R} = \frac{\partial_x^2 \ln(f_1(x))}{f_2(x)} + \frac{1}{2}\partial_x\left(\frac{1}{f_2(x)}\right)\partial_x \ln(f_1(x)). \tag{26}$$

**Anomalous temperature**  Let us now focus on the explicit expression of the energy-momentum tensor corrected by the gravitational anomalies $\mathcal{T} = \mathcal{T}_{\text{cl}} + \mathcal{T}_{\text{q}}$ with both components given in Eqns. (15,23). Using Eqns. (10,12,13), we obtain the expression for the density of energy, pressure, momentum and energy current:

$$\varepsilon = \frac{1}{2} C_w \left( \gamma T^2(x) + \varepsilon_q^{(1)} \right), \tag{27a}$$

$$p = \frac{1}{2} C_w \left( \gamma T^2(x) - \varepsilon_q^{(1)} \right), \tag{27b}$$

$$J_\varepsilon = v_F^2 \Pi = C_g \frac{\pi}{12\hbar} \, k_B^2 T^2(x). \tag{27c}$$

Remarkably, only two scales of energy set these values: the temperature $T(x)$, which incorporates $\varepsilon_q^{(2)}$ as we will see below in Eq. (28), and the quantum scale $\varepsilon_q^{(1)}$ defined in Eq. (24). This new scale $\varepsilon_q^{(1)}$ signals that in the presence of gravitational anomalies, the Stefan-Boltzmann law (9) is modified. The additive correction in (27a) signals a correction to the vacuum energy density at $T = 0$. This energy shift of pure geometrical origin is set by the local spacetime curvature $R$ as opposed to the analogous Casimir effect set by confinement [9]. It renormalizes pressure $p$ and energy density $\varepsilon$ in opposite directions.

The temperature $T^{-1} = ds/d\varepsilon$ is now set by the sum $\varepsilon + p = Ts = C_w \gamma T^2$, where $s$ denotes the entropy density. Equivalently, for each chiral branch of particles this temperature can be deduced from the off-diagonal components of the energy momentum tensor, the energy current and momentum in Eq. (27c). While the diagonal components of $\mathcal{T}$ including $\varepsilon + p$ are corrected by the trace anomaly, the off-diagonal components $J_\varepsilon$ and $\Pi$ are corrected by by the Einstein anomaly. Yet, the same temperature is defined consistently from both diagonal and off-diagonal quantities: both the trace and Einstein anomalies contribute coherently to correct the Tolman-Ehrenfest temperature into a generalized equilibrium temperature

$$\gamma T^2(x) = \gamma T_{\text{TE}}^2 + \varepsilon_q^{(2)}, \tag{28}$$

where the additive quantum correction $\varepsilon_q^{(2)}$ is defined in Eq. (24). Note that in defining this temperature, we restricted ourselves to the natural case where the entropy density $s$ is positive, which warrants that the right hand side of Eq. (28) is positive.

**Anomalous covariantly conserved momentum-energy tensor.**  As we discussed in Sec. 2.3, the Einstein anomaly can be traded for the Lorentz one, at the expense of a transformation (20) of the momentum-energy tensor. Such an expression describes the situation where the Lorentz invariance of quantum fluctuations is not enforced, as occurs naturally in condensed matter. The corresponding quantum correction to the momentum-energy tensor is now expressed as

$$[\tilde{\mathcal{T}}_q(x)]^\mu{}_\nu = \begin{pmatrix} \frac{C_w}{2} \left( \varepsilon_q^{(1)} + \varepsilon_q^{(2)} \right) & \frac{C_g}{2} \sqrt{\frac{f_1}{f_2}} \left( \varepsilon_q^{(2)} - \varepsilon_q^{(1)} \right) \\ -\frac{C_g}{2} \sqrt{\frac{f_2}{f_1}} \left( \varepsilon_q^{(1)} + \varepsilon_q^{(2)} \right) & \frac{C_w}{2} \left( \varepsilon_q^{(1)} - \varepsilon_q^{(2)} \right) \end{pmatrix}. \tag{29}$$

In this case, the momentum-energy tensor is no longer symmetric. As a consequence, the momentum density $\Pi$ and the density of energy current $J_\varepsilon$ are now distinct quantities. The chiral currents and momenta satisfy $J_{\varepsilon,\pm} = \pm v_F \, p_\pm$ and $\Pi_\pm = \pm \frac{1}{v_F} \, \varepsilon_\pm$ corresponding to the expressions

$$\varepsilon = \frac{1}{2} C_w \left( \gamma T^2(x) + \varepsilon_q^{(1)} \right), \tag{30a}$$

$$p = \frac{1}{2} C_w \left( \gamma T^2(x) - \varepsilon_q^{(1)} \right), \tag{30b}$$

$$v_F^{-1} J_\varepsilon = \frac{1}{2} C_g \left( \gamma T^2(x) - \varepsilon_q^{(1)} \right), \tag{30c}$$

$$v_F \Pi = \frac{1}{2} C_g \left( \gamma T^2(x) + \varepsilon_q^{(1)} \right). \tag{30d}$$

## 2.5 Anomalous Luttinger relation

The quantum anomaly correction to the Tolman-Ehrenfest relation Eq. (1) raises the question of whether this correction translates also to the Luttinger metric (5) with $f_1(x) = e^{2\phi}$ and $f_2 = 1$. While the connection between the gravitational anomaly and the energy current was discussed [61–63], its interplay with the Luttinger equivalence has been elusive so far. By inserting the Luttinger metric in Eqs. (18) and (25) we obtain

$$\varepsilon_q^{(1)} = \frac{\hbar v_F}{24\pi} \left[ \partial_x^2 \phi + (\partial_x \phi)^2 \right], \qquad \varepsilon_q^{(2)} = \frac{\hbar v_F}{24\pi} \partial_x^2 \phi, \tag{31}$$

corresponding, *via* Eq. (28), to a Luttinger temperature corrected by a second derivative of $\phi$:

$$\frac{T^2(x)}{T_0^2} = e^{-2\phi(x)} + \lambda_{T_0}^2 \partial_x^2 \phi, \quad \lambda_T = \frac{\hbar v_F}{2\pi k_B T}, \tag{32}$$

where $T_0$ is the reference temperature introduced Eqs. (1) and (11), and which is now chosen as $T_0 = T(x_0)$ at a point $x_0$ such that $\phi(x_0) = \partial_x^2 \phi(x_0) = 0$. Multiplying Eq. (32) by $T_0^2 e^{2\phi}$ and then differentiating with $x$ we obtain a correction to the original Luttinger relation (2) by an additional term induced by quantum anomalies:

$$\frac{\partial_x T}{T} = -\partial_x \phi + \lambda_T^2(x) \left[ (\partial_x \phi) \partial_x^2 \phi + \frac{1}{2} \partial_x^3 \phi \right]. \tag{33}$$

Notice that, since the equilibrium temperature $T(x)$ is inhomogeneous, the thermal length $\lambda_T$ is a coordinate-dependent quantity.

The energy density, pressure, energy currents and momentum are provided by Eqs. (27) or Eqs. (30). By using the explicit expression (31) for the corrections, we get

$$\varepsilon = \frac{C_w}{2} \left[ \gamma T_0^2 e^{-2\phi(x)} + \frac{\hbar v_F}{12\pi} \partial_x^2 \phi + \frac{\hbar v_F}{24\pi} (\partial_x \phi)^2 \right], \tag{34a}$$

$$p = \frac{C_w}{2} \left[ \gamma T_0^2 e^{-2\phi(x)} - \frac{\hbar v_F}{24\pi} (\partial_x \phi)^2 \right], \tag{34b}$$

$$J_\varepsilon = v_F^2 \Pi = C_g \left[ \frac{\pi}{12\hbar} (k_B T_0)^2 e^{-2\phi(x)} + \frac{\hbar v_F^2}{48\pi} \partial_x^2 \phi \right], \tag{34c}$$

when Lorentz invariance is enforced. Alternatively, if we relax Lorentz invariance while imposing diffeomorphism symmetry, the thermal current and momentum no longer identify, and are expressed as

$$J_\varepsilon = C_g \left[ \frac{\pi}{12\hbar} (k_B T_0)^2 e^{-2\phi(x)} - \frac{\hbar v_F^2}{48\pi} (\partial_x \phi)^2 \right], \tag{35a}$$

$$v_F^2 \Pi = C_g \left[ \frac{\pi}{12\hbar} (k_B T_0)^2 e^{-2\phi(x)} + \frac{\hbar v_F^2}{48\pi} \left( 2\partial_x^2 \phi + (\partial_x \phi)^2 \right) \right]. \tag{35b}$$

## 2.6 Anomalous potentials for a constant temperature profile

The non-linearity of the anomalous Luttinger relation (32) and (33) between temperature and the gravitational field $\phi$ has profound consequences. In particular, a fixed temperature profile $T(x)$ corresponds to a continuum of fields $\phi(x)$, contrarily to the case of the standard Luttinger relation (4) or (2). This leads to an additional freedom in the choice of $\phi$ for a given $T$ profile, typically imposed by additional boundary conditions. Let us illustrate this point by considering a constant temperature $T(x) = T_0$. The standard Luttinger relation (4) imposes a coordinate-independent gravitational potential corresponding to a constant $\phi = $ const. In contrast, the anomalous relation (33) allows a constant temperature to be realized, in a weak $\phi$ field limit, $|\phi| \ll 1$, by a class of dilation fields of the form:

$$\phi(x) = \phi_0 + \phi_+ e^{\sqrt{2}x/\lambda_{T_0}} + \phi_- e^{-\sqrt{2}x/\lambda_{T_0}}, \tag{36}$$

valid provided the anomalous corrections to Eq. (36) are small, subjected to the condition $|x| \ll \lambda_{T_0}$. The arbitrary coefficients $\phi_\pm$ highlight the degeneracy of the gravitational zero modes (36) which parametrizes the anomalous isothermal surfaces in the metric space. Note that in the case of a finite system with periodic boundary conditions, imposing the smoothness of the gravitational potential allows to recover a unique gravitational potential $\phi(x) = \phi_0 = const$.

In the case of an arbitrary metric and for a constant temperature profile $T(x) = T_0$, the anomaly-corrected Luttinger relation (33) translates into the following differential equation for the gravitational zero modes:

$$\partial_\xi^3 \phi + 2\partial_\xi \phi \ \partial_\xi^2 \phi - 2\partial_\xi \phi = 0, \tag{37}$$

where we introduced the rescaled coordinate $\xi = x/\lambda_{T_0}$. The third-order differential equation (37) on the gravitational zero modes possesses one trivial degeneracy corresponding to a global coordinate-independent shift of the gravitational potential $\phi \to \phi + \phi_0$, and two physically important degeneracies which label the space of possible zero modes. Each mode is labelled by the value of the first and second derivatives of the factor $\phi$ at a spatial reference point $x_0$. Hence a unique choice of field $\phi(x)$ for a given profile $T(x)$ require fixing these higher derivatives with a boundary condition, such as periodic boundary conditions on smooth fields.

## 3 Quantum atmosphere of a black hole

We start our discussion of physical consequences of the anomalous Tolman-Ehrenfest relation (28) by revisiting the Hawking radiation from a black hole. This corresponds to the generic situation of a spacetime background with a large curvature $R$ which induces large anomalous quantum corrections $\varepsilon_q^{(1)}$ and $\varepsilon_q^{(2)}$, that are even comparable with the classical Tolmann-Ehrenfest temperature. For this purpose we consider a generic black hole characterized by a metric (5) with $f_1 = 1/f_2 = f$. Such a metric encompasses both Schwarzschild black holes [64] for $f(x) = 1 - x_H/x$, with $x_H$ the black hole horizon, as well as evanescent Callan-Giddings-Harvey-Strominger (CGHS) black holes [65], initially introduced in the context of string theory [66], for $f(x) = 1 - \exp[-\alpha(x - x_H)]$. Generically, we consider a metric $f(x)$ which is asymptotically flat $\lim_{x\to\infty} f(x) = 1$ and vanishes linearly as $x$ approaches the event horizon $x_H$: $f(x \to x_H^+) \approx 2\kappa c^{-2}(x - x_H)$ where $\kappa$ is its surface gravity.

**Anomalous fluctuations and Hawking radiation.** We focus on the outgoing chiral flux of particles of velocity $v_F = c$. Their momentum energy tensor is given by Eqs. (8,23) with

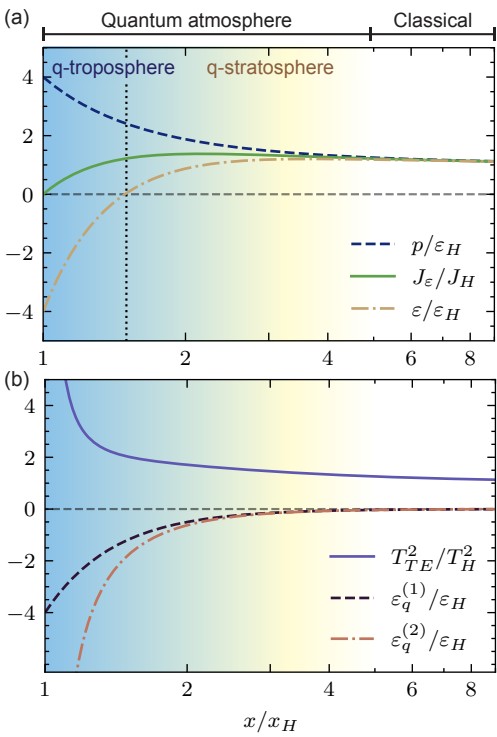

Figure 2: *Quantum atmosphere of a $D = 1 + 1$ Schwarzschild black hole.* (a) the dimensionless energy density $\varepsilon$, pressure $p$ and energy current $J_\varepsilon$, rescaled by their asymptotic values $\varepsilon_H = \frac{1}{2}\gamma T_H^2$ and $J_H = (\pi/12\hbar)k_B^2 T_H^2$ where $T_H$ is the Hawking temperature, are represented as a function of the distance $x$ to the center of the black hole, in units of its horizon $x_H$. Far from the horizon, all three quantities are proportional to $T_{\text{TE}}^2(x)$, where $T_{\text{TE}}(x)$ is the classical Tolman-Ehrenfest equilibrium temperature. Close to the horizon, quantum fluctuations strongly affect this classical behavior: an anomalous equilibrium temperature $T^2(x)$ is set by both $\varepsilon + p$ and $J_\varepsilon$. (b) The difference between the anomalous $T^2(x)$ and the classical $T_{\text{TE}}^2(x)$ is set by a quantum scale $\varepsilon_q^{(2)}$. The divergence of $T_{\text{TE}}^2$ at the horizon is counterbalanced by a diverging correction $\varepsilon_q^{(2)}$, leading to a vanishing $T^2(x)$. Simultaneously, the difference $\varepsilon_q^{(1)}$ between $\varepsilon$ and $p$ sets an independent quantum scale which remains finite at the horizon. These results illustrate the generation by a large spacetime curvature $R$ of a finite density of energy and asymptotic energy currents, which are captured by the trace and gravitational anomaly corrections to the thermodynamic quantities. In (a) and (b) The region where $T^2(x) \neq T_{\text{TE}}^2(x)$ defines the quantum atmosphere. Within it, we defined the quantum stratosphere, where $\varepsilon_q^{(1)} \approx \varepsilon_q^{(2)}$, and the quantum troposphere, where $\varepsilon_q^{(1)} \neq \varepsilon_q^{(2)}$, color-coding the smooth crossover between them. The vertical dotted line in (a) indicates where $\varepsilon = 0$.

$C_w = C_g = 1$. The two anomalous scales are deduced from Eqs. (18,24,25):

$$\varepsilon_q^{(1)} = \frac{\hbar c}{48\pi}\partial_x^2 f, \qquad \varepsilon_q^{(2)} = \frac{\hbar c}{48\pi}\left(\partial_x^2 f - \frac{(\partial_x f)^2}{2f}\right). \tag{38}$$

The corresponding thermal current, identical to the momentum, is deduced from Eqs. (27c,28), with a temperature $T(x)$ satisfying

$$k_B^2 T^2(x) = \frac{k_B^2 T_H^2}{f} + \frac{6\hbar c}{\pi}\varepsilon_q^{(2)}, \tag{39}$$

where we deduced $T_{\text{TE}}^2 = T_H^2/f(x)$ from Eq. (11) where $T_H$ is the asymptotic temperature at $x \to \infty$.

In both the Israel-Hartle-Hawking and Unruh vacua, the momentum tensor for the outgoing particles is regular at the horizon $x = x_H$ [67]. Therefore the divergence of $J_\varepsilon$ or $T^2(x)$ at the horizon, induced by the vanishing of $f(x \to x_H)$, has to be cancelled. The classical temperature $k_B^2 T_H^2/f$ always diverges at the horizon. On the other hand, the temperature (39) corrected by anomalous fluctuations remains finite at the horizon, provided we counterbalance the diverging classical temperature with the second contribution to $\varepsilon_q^{(2)}$ in Eq.(38). This amounts to imposing the condition

$$k_B T_H = \frac{\hbar}{2\pi c}\kappa\,. \tag{40}$$

The above reasoning demonstrates that anomalous fluctuations are essential close to the horizon given that the associated energy $\varepsilon_q^{(2)}$ corrects the spurious classical temperature divergence. Moreover, this subtle interplay between classical thermal and quantum fluctuations is at the origin of the asymptotic value of the temperature and energy variation. Quite remarkably, this implies that this radiation originates from these quantum fluctuations outside of the horizon. Indeed, plugging (40) into Eq. (39), we find that $T(x)$ and thus $J_{\varepsilon,+}$ vanish at the horizon [68], irrespective of the specific form of $f(x)$. No thermal current exits from inside the horizon. The asymptotic Hawking radiation $J_H = \frac{\pi}{12\hbar}k_B^2 T_H^2 = \hbar\kappa^2/(48\pi c^2)$ originates from a region *outside* of the black hole's horizon, its quantum atmosphere [28]. Let us now focus more closely on this region of strong anomalous fluctuations outside of the black hole.

**Quantum troposphere and stratosphere.** In Fig. 2 we illustrate the behavior of the energy density, pressure and energy current around the quantum atmosphere of a Schwarzschild black hole by choosing the metric $f(x) = 1 - x_H/x$. The thermodynamic quantities are rescaled by their asymptotic values $\varepsilon_H = \frac{1}{2}\gamma T_H^2$ and $J_H$ for $x \to \infty$. Between the horizon and $x \simeq 4x_H$ the effects of quantum fluctuations lead to sizable departure of $\varepsilon, p, J_\varepsilon, \Pi$ from their classical values. This is the quantum atmosphere of the black hole which hosts strong quantum fluctuations. Its extension depends on the specific black hole, corresponding to a choice of metric $f(x)$.

In the outer part of this atmosphere, anomalous corrections grow but the classical values still dominate: in particular the energy density remains positive. We denote this region the quantum stratosphere. Close to the horizon, irrespective of the choice of metric $f(x)$, the energy density $\varepsilon(x)$ always becomes negative. Indeed, from the decay of the gravity with $x$, we deduce that $\partial_x^2 f(x) < 0$, corresponding to a negative curvature $R$ in (24) and thus a negative scale $\varepsilon_q^{(1)}$ in (38). Given that $T(x)$ vanishes at the horizon, the energy density (27a) is negative close enough to the black hole, with an asymptotic value $\varepsilon = -p = (\hbar c/96\pi)\partial_x^2 f(x_H) < 0$.

This negative energy density is a hallmark of a region dominated by anomalous quantum fluctuations: classical fluctuations satisfy a Stefan-Boltzmann law (9) with an energy density always larger than that of the vacuum in flat spacetime $\varepsilon > 0$. We denote the region of $\varepsilon < 0$, where thermodynamic quantities are dominated by anomalous quantum fluctuations, the quantum troposphere.

Our analysis shows that the quantum atmosphere can be interpreted as the cradle of strong anomalous quantum fluctuations. In the quantum troposphere, the gravitational anomalies even dominate the thermodynamics. Signatures of such dominant quantum fluctuations are a negative energy density and large relative $\varepsilon - p$ compared to the average $\varepsilon + p$. In practice, the amplitude of the Hawking temperature is of the order of a few $10^{-8}$K for the smallest observed black holes [69], rendering the direct detection of these anomalous quantum phenomena elusive in real black hole. In the remaining of this paper, we will use the anomalous Luttinger

equivalence that we derive to explore analogues of the black hole quantum atmosphere occurring in situations where spacetime curvature is set by a temperature variations.

Before turning to these thermal analogues, let us briefly comment on historical references of the description of these anomalous quantum fluctuations. The relation between the Hawking radiation and quantum anomalies in $D = 1 + 1$ was pioneered by Christensen and Fulling who focused on the trace anomaly [25]. Robinson and Wilczek followed an alternative route by considering the consequences of the Einstein anomaly on an effective chiral theory [26,70,71]. The associated modified equilibrium temperature (28) was first derived in Ref. [72] while its relation to ballistic energy current (27c) through the Einstein anomaly was unnoticed. As we showed in section 2.4, both anomalies should be treated on the same footing when considering the effects of quantum fluctuations for a generic theory. The notion of quantum atmosphere of the a black hole, beyond its horizon, and at the origin of the Hawking radiation was recently discussed by Giddings [28] on CGHS black holes using conformal field theory techniques within the tortoise coordinates representation of the momentum energy tensor. This analysis was complemented in [27] by a general analysis of the Stefan-Boltzmann law accounting for the anomalous Tolman-Ehrenfest temperature.

## 4 Ballistic thermal response theory

The initial Luttinger relation (4) between a temperature profile and a gravitational potential has been instrumental in the description of thermal transport within linear response theory [1]. Having extended the Luttinger relation to incorporate anomalous quantum contributions, it is natural to explore its consequences on the linear and non-linear [73] response theory of thermal transport. This is further motivated by concerns [74, 75] on the applicability of a Green-Kubo approach of heat transport for large temperature gradients, questioning the equivalence between temperature and gravitational analogs [76].

Our goal in this section is to show that the anomalous relation (32) allows to relate the out-of-equilibrium energy current of Dirac fermions in an inhomogeneous temperature profile $T(x)$ to the energy current of Dirac fermions in a curved spacetime at finite homogeneous temperature. This curved spacetime is defined by a Luttinger metric of the form (3) with a gravitational potential $\phi_{an}[T(x)]$ satisfying the anomalous relation (33).

Following this route, the out-of-equilibrium energy current is determined within linear response theory in the small field $\phi_{an}$. Because (33) is nonlinear, one may wonder to what extent a perturbative response theory to linear order in $\phi_{an}$ can capture non-linear effects of temperature gradients. This section addresses this question.

In the following, we consider a ballistic conductor: by definition, chiral particles are not allowed to exchange energy with each other. Hence the energy density, current, pressure and momentum decompose into contributions of these left and right movers: each chiral species is considered independently from the other. For simplicity, we thus focus on a single chirality, *e.g.* right movers. Such a chiral conductor is realized either as a single component of a ballistic non-chiral conductor, or the edge channel of a Chern insulator such as a quantum Hall phase, represented in Fig. 1b. These right moving particles experience a local temperature $T_+(x)$. At equilibrium, $T_+(x) = T_0$, leading to a steady energy current and momentum $J_{\varepsilon,+}^{(0)} = v_F^2 \Pi_+^{(0)} = c_+ \frac{\pi}{12\hbar}(k_B T_0)^2$.

To drive the system out-of-equilibrium, we typically heat one side of the conductor. Considering a conductor if size $L$ extending from $x = -L/2$ to $x = L/2$, we set $T(-L/2) = T_L$. Allowing these energy carriers to exchange energy with a bath of phonons of the material, we expect their temperature to be inhomogeneous:

$$T(x) = T_0(1 + a \, x/L), \tag{41}$$

where the quantity $a = L\partial_x T/T_0$ is typically set by the rate of energy exchange between the carrier (electrons) and the phonons. This inhomogeneous temperature induces an excess current $J_{\varepsilon,+} - J_{\varepsilon,+}^{(0)}$ that we study in lowest orders in $a$.

## 4.1 Kubo formula and gravitational anomalies

We start by demonstrating the equivalence between the expression (35) for the energy current and that obtained by direct calculation within response theory linear in the gravitational potential $\phi$ . Hence, we consider a $D = 1+1$ chiral Dirac Hamiltonian in curved space is given by

$$\mathcal{H} = \int \mathrm{d}x \, e^{\phi(x)} \, \hat{h}_+(x), \tag{42}$$

where, in terms of second quantized fields $\Psi_+(x), \Psi_+^\dagger(x)$, the Hamiltonian density operator is

$$\hat{h}_+(x) = -\frac{i\hbar v_F}{2} \, \Psi_+^\dagger(x) \overleftrightarrow{\partial}_x \Psi_+(x). \tag{43}$$

where $\overleftrightarrow{\partial}_x = \overrightarrow{\partial}_x - \overleftarrow{\partial}_x$. From Eqs. (35), we learned that energy density and momentum operators have to be treated separately when quantum fluctuations are accounted for. Indeed, the momentum $\Pi_+ = -\frac{i\hbar}{2}\langle \Psi_+^\dagger \overleftrightarrow{\partial}_x \Psi_+ \rangle = \frac{1}{v_F}\varepsilon_+$ identifies with the energy current $J_{\varepsilon,+} = \frac{i\hbar v_F}{2}e^{-\phi(x)}\langle \Psi_+^\dagger \overleftrightarrow{\partial}_t \Psi_+ \rangle$ only for classical fields satisfying the equation of motion.

For simplicity we focus on the momentum density which only involves the equal time Green's function:

$$\Pi_+(x) = \int \frac{\mathrm{d}k\mathrm{d}q}{(2\pi)^2} \, e^{iqx} \left\langle \Psi_{+,k-\frac{q}{2}}^\dagger \hbar \, k \Psi_{+,k+\frac{q}{2}} \right\rangle = -i \int \frac{\mathrm{d}k\mathrm{d}q}{(2\pi)^2} \frac{\mathrm{d}\omega}{2\pi} e^{iqx} \hbar \, k G_{k+\frac{q}{2},k-\frac{q}{2}}^<(\omega), \tag{44}$$

where the lesser green functions $G^<$ is defined by

$$G_{k,k'}^<(\omega) = i \int_0^\infty \mathrm{d}t \, e^{i\omega t} \langle \Psi_{+,k}^\dagger(0)\Psi_{+,k'}(t) \rangle. \tag{45}$$

Working in perturbation theory at first order in the gravitational potential $\phi(x)$, we expand the Green function using the Dyson equation

$$G_{k',k}^<(\omega) = \left(G_0^<\right)_{k',k} + \int \mathrm{d}x \, e^{i(k-k')x}\phi(x)\left(\left(G_0^R\right)_{k',k'} h_{k',k}\left(G_0^<\right)_{k,k} + \left(G_0^<\right)_{k',k'} h_{k',k}\left(G_0^A\right)_{k,k}\right), \tag{46}$$

expressed in terms of the retarded Green's functions in flat spacetime

$$G_{k,k'}^R(\omega) = -i \int_0^\infty \mathrm{d}t \, e^{i\omega t} \langle \{\Psi_k^\dagger(0), \Psi_{k'}(t)\} \rangle. \tag{47}$$

the advanced Green's function $G_{k,k'}^A(\omega) = (G_{k,k'}^R(\omega))^*$ and the lesser Green's function (45). The equilibrium energy current density, corresponding to the the term of $0^{\text{th}}$ order in $\phi$, is

$$\Pi_+^{(0)}(x) = \overline{\Pi}_+ + \frac{\pi}{12\hbar v_F^2}k_B^2 T_0^2 + \mathcal{O}\left((k_B T_0)^4\right), \tag{48}$$

where $\overline{\Pi}_+$ is the Fermi sea contribution to the momentum. We obtain a first order in $\phi$ correction to this momentum leading to the expression

$$\Pi_+(x) = \overline{\Pi}_+ + \frac{\pi}{12\hbar v_F^2}k_B^2 T_0^2(1-2\phi) + \frac{\hbar}{24\pi}\partial_x^2\phi(x) + \mathcal{O}\left((k_B T_0)^4\right). \tag{49}$$

This expression identifies with (35b) to lowest order in $\phi$ and its derivatives. This demonstrates that gravitational anomalies corrections to the expression of the energy current and momentum are captured by a standard Kubo-Green expansion perturbative in the gravitational potential $\phi$.

### 4.2 From a temperature profile to an equivalent gravitational potential: Nonlinear thermal conductivity

Having established the equivalence between the Eqs. (35) and standard linear in $\phi$ response theory, we need to express these relations in terms of the temperature profile $T(x)$. This aim is achieved by solving the equation (32) perturbatively in $a = L\partial_x T/T_0$ for the temperature profile (41) (see Appendix D). We obtain a gravitational potential

$$\phi_{\text{an}}[T(x)] = \phi_{\text{Lutt}} + \delta\phi, \tag{50}$$

where $\phi_{\text{Lutt}} = \ln(T_0/T(x))$ is the potential deduced from the standard Luttinger equivalence (4), and $\delta\phi$ encodes the corrections induced by the gravitational anomalies:

$$\delta\phi = a^2\frac{\lambda_{T_0}^2}{2L^2} - 2a^3\frac{\lambda_{T_0}^2}{L^2}\frac{x}{L} + \mathcal{O}(a^4). \tag{51}$$

We now express the chiral energy current and momentum of chiral Dirac fermions in a temperature profile $T(x)$ using the expression (50,51) of the equivalent gravitational potential. From Eqs. (35) they identify with $v_F^2\Pi_+/J_0 = T^2/T_0^2 + \varepsilon_q^{(1)}/(\gamma T_0^2)$; $J_{\varepsilon,+}/J_0 = T^2/T_0^2 - \varepsilon_q^{(1)}/(\gamma T_0^2)$ with $J_0 = v_F\gamma T_0^2/2$. The quantum anomalies corrections are encoded solely in $\varepsilon_q^{(1)}$. which induce a modification of the the thermal current at non-linear order in the temperature gradient $a = L\partial_x T/T_0$. At this stage, we realize that $\delta\phi$ of Eq. (51) are at least of order $(a/L)^4$. Hence, to second order in the temperature gradient, we can indeed neglect the modification of this gravitational potential due to the anomalous Luttinger relation. Inserting the bare Luttinger potential $\phi_{\text{Lutt}}$ in the expression (31) $\varepsilon_q^{(1)}$, we obtain the following expression for the current

$$\frac{J_{\varepsilon,+}}{J_0} = 1 + 2x\,\frac{\partial_x T}{T_0} + \left[x^2 + 4\lambda_{T_0}^2\right]\left(\frac{\partial_x T}{T_0}\right)^2 + \mathcal{O}\left(\frac{\partial_x T}{T_0}\right)^3, \tag{52}$$

with a thermal lengthscale $\lambda_{T_0}$ defined in (32) at the reference temperature $T_0$.

This expression encodes the effects of the gravitational anomaly within a regime of small thermal gradients. Note that it is non-linear in thermal gradient $\nabla T/T_0$, although it originates from an expression linear in $\phi$ and its derivative. This illustrates that linear response theory in the gravitational potential $\phi$ can apply beyond the regime linear in $\nabla T/T_0$. In the remaining section of this paper, we will focus on consequences of the quantum anomalies on transport beyond this regime of small temperature gradient, far from equilibrium, where we expect their effects to be even more pronounced.

## 5 Far from equilibrium energy transport

A quench procedure, in which external parameters controlling an equilibrium system are suddenly changed, allows to explore dynamics of quantum systems beyond the realm of linear response theory. The rich possibilities offered by experiments using ultra-cold atoms have triggered a recent interest in such an out-of-equilibrium dynamics [77].

In this section, we focus on the situation of a finite size quantum system connected to a thermal bath, whose temperature is varied rapidly. In a standard partition procedure, two halves

of the conductor are maintained at different temperatures $T_{R/L} = T_0 \mp \Delta T/2$, see Fig. 3(a). The corresponding external temperature profile $T(x)$ maintains the conductor in an out-of-equilibrium state. The temperature profile is then later released at $t = 0$: in the equilibration process between different regions of the conductor, local heat currents appear. Remarkably, an oscillating heat wave was observed in a pioneering numerical study on spin chains [32], later described analytically [29, 31].

In this section, we demonstrate that these oscillations, as well as an associated pressure discontinuity at time $t = 0$, are measurable signatures of the gravitational and trace anomalies. They originate from the energy density characterizing the steady out-of-equilibrium state at time $t < 0$ which we describe first. In a later refinement of this quench physics, we consider Floquet states generated by periodically imposing and releasing an external temperature profile.

## 5.1 Anomalous Luttinger relation on a ring

### 5.1.1 From generalized Gibbs measure to curved spacetime

We consider a generic interacting gas on a ring of size $L$, described at low energy by a relativistic Luttinger liquid [78–82]. For time $t < 0$, this system is spatially modulated either by a variation of the interactions or by an external temperature. In the resulting inhomogeneous out-of-equilibrium steady state, physical observables $\langle \mathcal{O} \rangle$ are assumed to be described by statistical averages with a generalized Gibbs measure

$$\langle \mathcal{O} \rangle = \frac{\text{Tr } \mathcal{O} \, e^{-\mathcal{G}}}{\text{Tr } e^{-\mathcal{G}}}, \quad \mathcal{G} = \int_0^L dx \, \frac{1}{k_B T_0 \xi(x)} \mathcal{H}(x), \tag{53}$$

where $\mathcal{H}(x)$ is the Hamiltonian density and $\xi(x)$ the parameter of the spatial modulation. It is natural to expect that the local equilibrium temperature of the wire is set by

$$T(x) \overset{?}{=} T_{\text{TE}}(x) = \xi(x) T_0 \,. \tag{54}$$

However, we show below that this is not the case. To engineer a given temperature profile, gravitational anomalies corrections have to be accounted for to determine the equivalent profile $\xi(x)$. Besides, our results demonstrates that equivalence between modulating the velocity or the inverse temperature of relativistic excitations require some particular care.

To identify the local equilibrium temperature corresponding to the generalized Gibbs measure (53), we start by interpreting it as a Gibbs measure at constant temperature $T_0$ but in a curved spacetime with the metric[2] (6) associated to the Luttinger gravitational potential $\phi_{\text{Lutt}}(x) = -\ln \xi(x)$:

$$\mathcal{G} = \frac{1}{k_B T_0} \int_0^L dx \sqrt{f_1} \mathcal{H}(x) \,. \tag{55}$$

We can now use our results of section 2.4: the equilibrium temperature $T(x)$ in this curved spacetime does not identify with the standard Tolman-Ehrenfest $T_{\text{TE}}(x)$: the difference is a direct measure of the amplitude of the corrections due to the trace and gravitational anomalies. More precisely, let us recall the relation (28): $\gamma T^2(x) = \gamma T_{\text{TE}}^2 + \varepsilon_q^{(2)}$ where the quantum energy scale

$$\varepsilon_q^{(2)} = \frac{\hbar v_F}{24\pi} \left[ -\frac{\partial_x^2 \xi}{\xi} + \left( \frac{\partial_x \xi}{\xi} \right)^2 \right] = \frac{\hbar v_F}{24\pi \ell_T^2} \,, \tag{56}$$

---

[2]We should be careful to express thermodynamic quantities in the laboratory frame when using the curved spacetime representation.

depends on the length $\ell_T$ which encodes the local variation of the metric:

$$\ell_T(x) = \left|\partial_x^2 \ln \xi(x)\right|^{-\frac{1}{2}} . \tag{57}$$

The relative correction to the temperature is thus set by a ratio of lengths:

$$\frac{T^2(x)}{T_{\text{TE}}^2(x)} = 1 + \left(\frac{\lambda_T(x)}{\ell_T(x)}\right)^2 , \tag{58}$$

where the thermal length $\lambda_T = \hbar v_F / (2\pi k_B T_{\text{TE}}(x))$, see Eq. (32).

From Eqs. (34) we obtain the energy density and pressure[3]

$$\varepsilon = \frac{1}{2}(c_+ + c_-)\left(\gamma T^2 + \varepsilon_q^{(1)}\right) , \tag{59}$$

$$p = \frac{1}{2}(c_+ + c_-)\left(\gamma T^2 - \varepsilon_q^{(1)}\right) . \tag{60}$$

with the anomalous quantum scale

$$\varepsilon_q^{(1)} = \frac{\hbar v_F}{24\pi}\left[-\frac{\partial_x^2 \xi}{\xi} + 2\left(\frac{\partial_x \xi}{\xi}\right)^2\right] = \frac{\hbar v_F}{24\pi}\left[\ell_T^{-2} + \tilde{\ell}_T^{-2}\right] , \tag{61}$$

whose amplitude is set by both the length $\ell_T$ from Eq. (57) and a second length scale parametrizing temperature variations:

$$\tilde{\ell}_T(x) = \left|\frac{\xi}{\partial_x \xi}\right| . \tag{62}$$

Similarly, the energy current and momentum densities read

$$J_{\varepsilon,\pm} = \pm\frac{v_F}{2}c_\pm\left(\gamma T^2 - \varepsilon_q^{(2)}\right) = \pm c_\pm\left[\frac{\pi}{12\hbar}(k_B T_0)^2 \xi^2 - \frac{\hbar v_F^2}{48\pi}\left(\frac{\partial_x \xi}{\xi}\right)^2\right] , \tag{63}$$

$$\Pi_\pm = \pm\frac{1}{2v_F}c_\pm\left(\gamma T^2 + \varepsilon_q^{(1)}\right) = \pm c_\pm\left[\frac{\pi}{12\hbar v_F^2}(k_B T_0)^2 \xi^2 + \frac{\hbar}{48\pi}\left(3\left(\frac{\partial_x \xi}{\xi}\right)^2 - 2\frac{\partial_x^2 \xi}{\xi}\right)\right] . \tag{64}$$

### 5.1.2 Inhomogeneous temperature

Let us evaluate the amplitude of the corrections by quantum fluctuations encoded in the trace and gravitational anomalies by considering a non-chiral wire with central charges $c_+ = c_- = 1$, maintained in an out-of-equilibrium state by a temperature profile $T(x)$. This temperature is constant in two regions with values $T_{R/L} = T_0 \mp \Delta T/2$, and smoothly interpolates over a length $\delta$ between them, at positions $x = 0, L/2$ as displayed in Fig. 3(a). Although our approach applies to a generic temperature profile, for the sake of clarity we choose a profile:

$$T(x) = T_0 - \frac{\Delta T}{2}\,\tanh\left[\frac{L}{2\pi\delta}\sin\left(2\pi\frac{x}{L}\right)\right] . \tag{65}$$

Given this temperature profile, we identify the equivalent energy density modulation $\xi(x)$ by inverting numerically the relation (58). This functions $\xi(x)$ is then used to calculate the amplitudes of the quantum corrections $\varepsilon_q^{(1)}$ and $\varepsilon_q^{(2)}$ and the corresponding densities and current. The results are shown on Fig. 3. We expect the gravitational anomalies to alter the

---

[3]Here, and in the following, we neglect an additive finite size correction [83] $\varepsilon_C = -\pi\hbar v_F/(24L^2)$ to both $\varepsilon$ and $p$. We consider situations where this correction is negligible with respect to $\varepsilon_q^{(1)}, \varepsilon_q^{(2)}$.

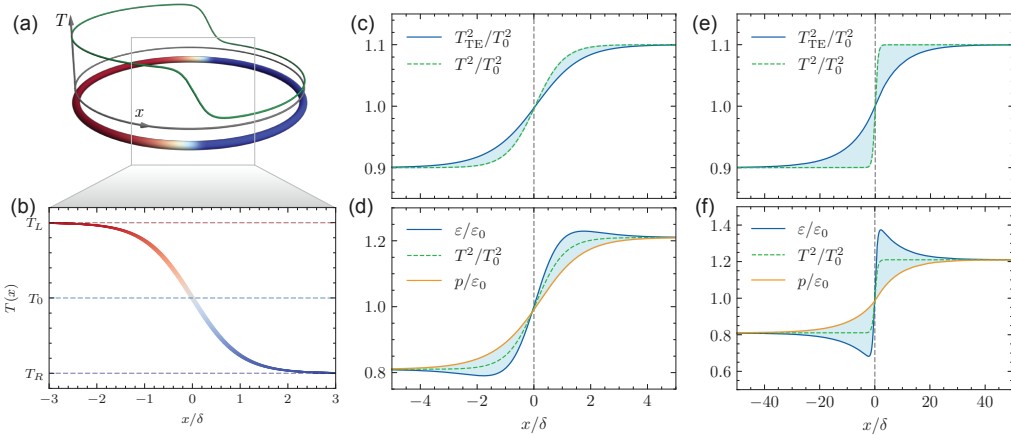

Figure 3: *Quantum corrections to the out-of-equilibrium steady state imposed by a temperature jump.* (a) Two halves of a close ring of non-interacting particles ($c_+ = c_- = 1$) are set at two temperatures $T_{R/L} = T_0 \mp \Delta T/2$. (b) At the two contacts, the temperature smoothly varies over a region of size $\delta$. We consider a Fermi velocity $v_F = 10^6 \text{m} \cdot \text{s}^{-1}$ typical for relativistic materials, a cryogenic temperature $T_0 = 100$ mK and a small relative temperature jump $\Delta T/T_0 = 0.2$. (c) and (d) for a size of the contact region $\delta = 10 \, \mu\text{m}$, the pressure, shown relative to its means value $\varepsilon_0 = \gamma T_0^2$, follows the classical law $p = \gamma T^2$. On the other hand the energy density $\varepsilon(x)$ departs from this classical law: the amplitude of the corresponding corrections, represented by the shaded area, are set by a quantum energy scale $\varepsilon_q^{(1)}$. This energy scale originates from quantum fluctuations, at the origin of scale and gravitational anomalies, or similar origin that that in the black hole's atmosphere. In the present case the curvature $R$ of spacetime is set by the imposed temperature through the Luttinger equivalence relation. (e) and (f) for a smaller size $\delta = 1 \, \mu\text{m}$, two quantum energy scales $\varepsilon_q^{(1)}$ and $\varepsilon_q^{(2)}$ have to be distinguished. While $(\varepsilon_q^{(1)} + \varepsilon_q^{(2)})/2$ still appears as the amplitude of the oscillating corrections to $\varepsilon$, the difference $\varepsilon_q^{(1)} - \varepsilon_q^{(2)}$ manifests itself both in the asymmetry of these corrections around the temperature jump, as well as a departure of the pressure from the classical law.

classical properties of the steady state in regions where $\lambda_T(x) \lesssim \ell_T(x)$, *i.e.* close to the temperature jumps for strong enough relative variation of this temperature. Therefore we focus on the temperature jump around $x = 0$ of the temperature profile (65), as shown in Fig. 3(b). For a steady state, $J_\varepsilon$ and $\Pi$ vanish.

The parameters of Fig. 3 are motivated by relativistic electronic conductors. In graphene [84], Carbon nanotubes [85] and Dirac and Weyl semimetals [86], the Fermi velocity of Dirac particles is of the order $v_F \sim 10^6 \text{ ms}^{-1}$, yielding a thermal length $\lambda_{T_0} \times T_0 \simeq 7.64 \times 10^{-6}$ K $\times$m for a dilution refrigerator temperature of $T_0 = 100 \text{ mK}$. We choose a relative temperature jump $\Delta \xi = 0.2$. For smooth temperature jump over a length $\delta = 10 \mu\text{m}$, we obtain from (58) $\Delta \xi \ll 1$: $\tilde{\ell}_T(x)$ is very large and $\ell_T(x) \ll \tilde{\ell}_T(x)$. A single length scale $\ell_T(x) \simeq |\xi/\partial_x^2 \xi|^{1/2}$, set by the Ricci scalar $R$, characterizes the anomalous fluctuations. Correspondingly, gravitational anomaly corrections involve a single quantum energy scale $\varepsilon_q^{(1)} \approx \varepsilon_q^{(2)} \approx -\hbar v_F/(24\pi \ell_T^2)$. Both the pressure and the energy density display small departures from the classical law $\varepsilon = p = \frac{1}{2}(c_+ + c_-)\gamma T^2$, as shown in Fig. 3(d). The amplitude of this correction, symmetric around the temperature jump, corresponding to the shaded area, is a direct measure of the quantum correction $\varepsilon_q^{(1)}$ set by the anomalies. For sharper temperature jump over $\delta = 1\mu\text{m}$, we observe that the corresponding Gibbs or Tolman-Ehrenfest temperature $T_{\text{TE}}(x)$ is much smoother, as shown in Fig. 3(e). This illustrates that an inhomogeneous temperature induces

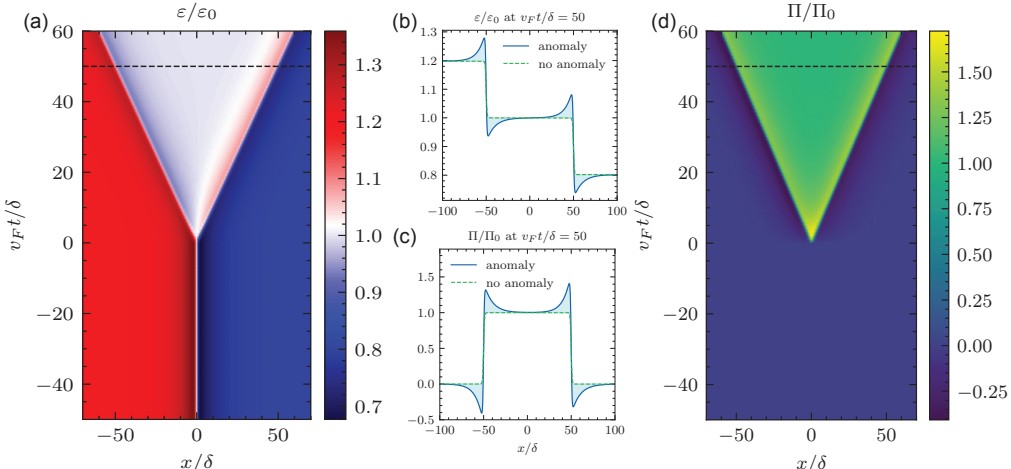

Figure 4: *Quantum corrections to energy traveling waves imposed by a temperature quench.* (a) Two halves of a wire of non-interacting particles ($c_+ = c_- = 1$) with Fermi velocity of $v_F = 10^6$ ms$^{-1}$ are set at two temperatures $T_{R/L} = T_0 \pm \Delta T/2$ following the same protocol as in Fig. 3. The average temperature is $T_0 = 100$mK, and the ramp of temperature of amplitude $\Delta T = 20$mK is imposed over a length $\delta = 1\mu$m. At time $t = 0$ this external temperature difference is released. Following this quench, two traveling waves of energy appear. (b) The non-monotonous behavior of the density of energy profile is a manifestation of the anomaly corrections originating from quantum fluctuations of similar nature than close to a black hole. The amplitude of the corrections, represented by the shaded area, is a direct measure of the new quantum scale of energy $\varepsilon_q^{(1)} \simeq \varepsilon_q^{(2)}$ set by gravitational anomalies. In between the two waves, appears a region of homogeneous density of energy $\bar{\varepsilon} = \frac{1}{2}\gamma(c_+ T_L^2 + c_- T_R^2) = \gamma(T_0^2 + \Delta T^2)$. (c) and (d) This intermediate region is not in equilibrium: it is crossed by right moving particles at temperature $T_L$ and left moving particles at temperature $T_R$, leading to a steady current $\bar{J}_\varepsilon = \frac{1}{2}\gamma v_F(c_+ T_L^2 - c_- T_R^2) = 2\gamma v_F T_0 \Delta T$ and momentum $\bar{\Pi} = v_F^{-2} \bar{J}_\varepsilon$. The oscillating corrections to the momentum or energy current close to the interface between the three regions, shown in (b) and (c) as shaded areas, are also an accessible manifestation of corrections due to quantum fluctuations due to the trace and gravitational anomalies, similarly to those at the vicinity of a black hole. In the present case they originate from the local strong curvature of the effective spacetime accounting, following Luttinger equivalence, for the temperature variation.

analog gravitational potentials that are smoother than those induced by variations of the velocity, *e.g.* by varying interactions. Remarkably the energy density display some deep and spike around the temperature jump, represented in Fig. 3(e), which are signatures of the gravitational anomaly corrections. In that situation, the two lengthscales $\ell_{T_{\text{ext}}}(x)$ and $\tilde{\ell}_{T_{\text{ext}}}(x)$ slightly differ, corresponding to two different quantum energy scales $\varepsilon_q^{(1)}$ and $\varepsilon_q^{(2)}$. However, in practice, only $\varepsilon_q^{(1)}$ leads to experimentally measurable corrections through the difference $\varepsilon - p = (c_+ + c_-)\varepsilon_q^{(1)}$.

## 5.2 Temperature quench as a metric quench

In practice, maintaining a conductor in a steady out-of-equilibrium state is difficult and not practical: it is often easier to study the dynamics following a corresponding quench. As we

show below, the dynamics reflects the quantum corrections to the initial steady state. Thus we consider a situation where the the temperature profile (65) is imposed up to time $t = 0$, and released afterwards.

The out-of-equilibrium dynamics occurs in a closed system, but a ballistic evolution forbids exchange of energy between left and right movers. Given that the equilibrium temperature of such a system is uniform, the Tolman-Ehrenfest equivalence implies that this dynamics occurs in a flat spacetime, with vanishing curvatures $R = 0$ and $\bar{R} = 0$. In this flat spacetime, $\varepsilon = p$ and $v_F^2 \Pi = J_\varepsilon$.

Following the extended Luttinger correspondence that we developed in section 5.1, the out-of-equilibrium dynamics at time $t > 0$ can be viewed as resulting from a quench of the spacetime metric at $t = 0$ from the Luttinger metric to a flat metric. Continuity conditions on the momentum energy tensor, derived in Appendix F, imply that both the energy density $\varepsilon$ as well as the momentum $\Pi$ are continuous during this quench of metric: $\varepsilon(t = 0^+) = \varepsilon(t = 0^-)$ and $\Pi_\pm(t = 0^+) = \Pi_\pm(t = 0^-)$. On the other hand the energy current density is discontinuous, with $J_\pm^\varepsilon(t = 0^+) - J_\pm^\varepsilon(t = 0^-) = c_\pm v_F \varepsilon_q^{(1)}$ resulting in a pressure discontinuity $\Delta p = (c_+ + c_-)\varepsilon_q^{(1)}$, where $\varepsilon_q^{(1)}$ is set by (61).

Given that low-energy excitations of our system evolve ballistically, we obtain for time $t > 0$ $J_\pm^\varepsilon(x, t) = v_F^2 \Pi_\pm(x, t) = v_F^2 \Pi_\pm(x \mp v_F t, 0^+)$ where the momenta at $t = 0^+$ are defined in (64). The resulting energy density and momentum are represented in Fig. 4 for the same parameters than Fig. 3(e) and (f). The quantum corrections characterizing the energy density and pressure of the steady state at $t < 0$ now manifests themselves as traveling waves of energy after the quench, as shown in Fig. 4(a) and (b). In between the two traveling waves emerges a region of homogeneous density of energy

$$\bar{\varepsilon} = \frac{1}{2}\gamma\left(c_+ T_L^2 + c_- T_R^2\right) = \gamma\left(T_0^2 + \Delta T^2/4\right), \tag{66}$$

where the average temperature $T_0$ is defined in Fig. 3(b). In this region, right moving particles carry an energy density $\frac{1}{2}c_+\gamma T_L^2$ while left moving particles carry an energy density $\frac{1}{2}c_-\gamma T_R^2$, resulting in a steady-state value of the current $\bar{J}_\varepsilon = \frac{1}{2}\gamma v_F(c_+ T_L^2 - c_- T_R^2) = 2\gamma v_F T_0 \Delta T$ and momentum $\bar{\Pi} = v_F^{-2}\bar{J}_\varepsilon$. This expression agrees with the pioneering study on interacting chains [32] and [49] as well as a Landauer-Büttiker approach for non-interacting fermions [87].

The traveling waves of energy, shown in Fig. 4(a) and (b), reflect as traveling waves of momentum shown in Fig. 4(c) and (d). The amplitudes of the quantum corrections, represented as the shaded area, is equal to that of the energy density before the quench, shown in Fig. 3: it is set by $(\varepsilon_q^{(1)} + \varepsilon_q^{(2)})/2$ where the two scales are defined in (61,56). The asymmetry of these corrections around the wave front is set by $\varepsilon_q^{(1)} - \varepsilon_q^{(2)}$.

These results, which we derived from gravitational anomaly corrections in a curved spacetime as well as continuity conditions following a metric quench, were previously derived using series expansions and conformal field theory techniques in [29, 31]. The presence of a Schwarzian derivative in the expression for the density of energy and energy current can indeed be traced back to a manifestation of the trace anomaly identified in the present paper. Through the (extended) Luttinger equivalence, a thermal quench can be treated as a quench of metric which appears seemingly identical to a quench imposed by the release of an external confining potential considered in [88]. In both case, anomalies capture the quantum corrections induced by large spacetime curvatures.

## 5.3 Spacetime periodic modulation: Floquet states

In this section, building on the above study of single thermal quench, we explore how anomalous quantum fluctuations appear on periodic sequences of quantum quenches. While periodic

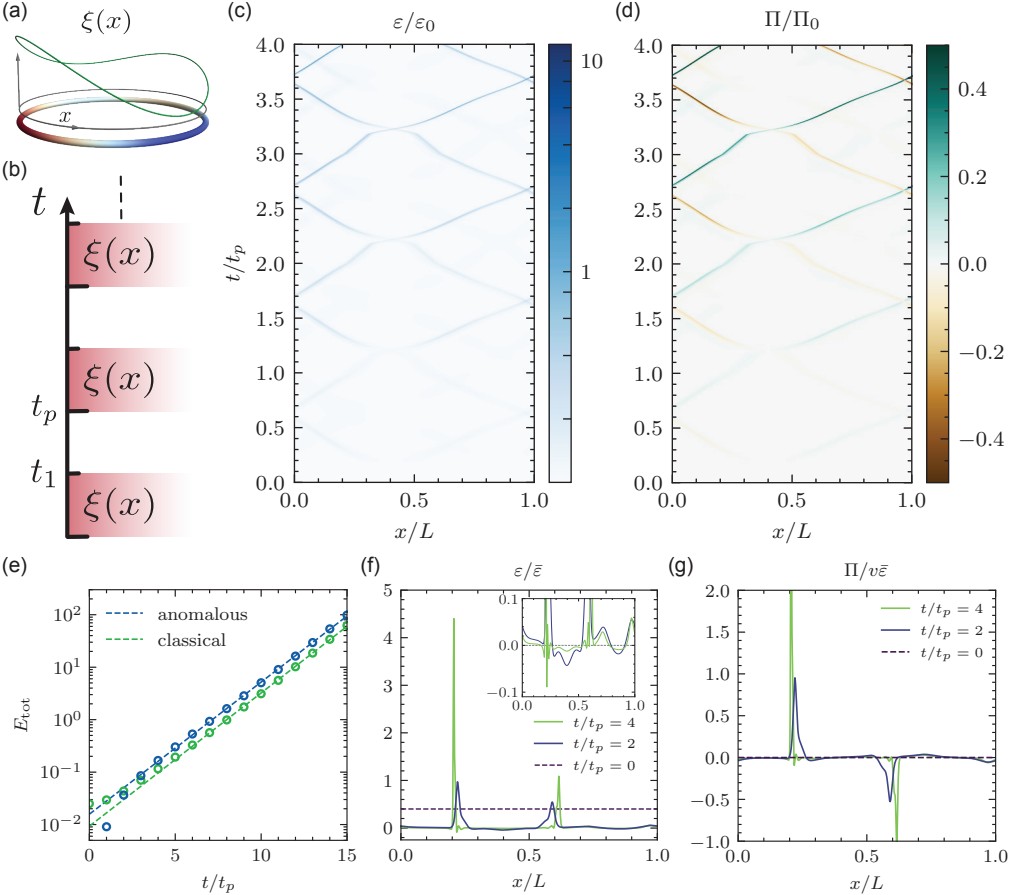

Figure 5: *Floquet heating state.* (a) and (b) We consider a ring of free relativistic fermions with a spatially modulated velocity $\xi(x)v_F = 1$. The modulation $\xi(x)$ is periodic in time with a period $t_p = t_1 + t_2$, such that (i) during time $t_1$ set to the smooth profile shown in (a), and (ii) during time $t_2$ no modulation is applied and $\xi(x) = 1, \forall x$. The two times $Lt_1/v_F = 0.1$ and $Lt_2/v_F = 0.45$, are chosen such that the period coincides with the time of flight of particles around the ring: $L = v_F t_1 + \tilde{v}_F t_2$ where $\tilde{v}_F$ is the averaged effective velocity over the profile $\xi(x)$. (c) and (d) As a function of time, both the energy density $\varepsilon$ and the momentum $\Pi$ become highly inhomogeneous, and concentrate on a few trajectories. They are represented rescaled by the classical values $\varepsilon_0 = v_F \Pi_0 = \gamma T_0^2$. The energy and momentum profiles are represented after two and four periods in panels (f) and (g), illustrating the localization mechanism. (e) Besides being focused spatially, the net energy of the ring $E_{\text{tot}} = \int_0^L \varepsilon \, dx = L\bar{\varepsilon}$ increases: the Floquet state is heating. This is represented by monitoring the stroboscopic dynamics at times $t_n = nt_p$ of the ring for which $E_{\text{tot}}$ increases exponentially. Remarkably the rate of increase of this energy is not classical: quantum fluctuations, responsible for the trace and gravitational anomaly corrections, have a growing energy. The two focusing trajectories behave as heating black holes: in their neighborhood the energy density becomes negative, as shown in the inset of panel (f). This is an additional manifestation of the effects of quantum fluctuations induced by a local large curvature similar to those in a black-hole atmosphere.

thermal quenches realize the same physics, for technical reasons we follow the protocol recently proposed in [35–38] by implementing directly a quench of metric $\xi(x)$, see Eq. (53). This choice allows to bypass the numerical determination of a metric equivalent to a thermal profile as done in section 5.1. By interpreting time-periodic, or Floquet, change in the spatial dependence of the system's parameters as metric quench (Fig. 5(a)), we highlight the role that gravitational and trace anomalies play in the phenomenology of the resulting Floquet conformal field theories [33, 34].

The peculiarity of Floquet conformal field theories relies on the striking, but analytic, thermalization properties [35–38] occurring when periodically modulating the system between two inhomogeneous states. This two-step periodic drive is obtained when the dynamics of particles on a circle of size $L$ is alternatively described by a uniform and an inhomogeneous Hamiltonian:

$$\mathcal{H} = \int_0^L dx \, \frac{1}{\xi(x,t)} h_0(x), \tag{67}$$

where

$$\xi(x,t) = \begin{cases} \xi(x), & \text{for } t \in [0, t_1], \\ 1, & \text{for } t \in [t_1, t_1 + t_2 \equiv t_p], \end{cases} \tag{68}$$

where $t_p$ is the period. While initially $\xi(x)$ was chosen to be an inverse sine squared deformation $\xi^{-1}(x) = 2\sin^2(\pi x/L)$ [33, 34], we consider more general profiles in the following [37, 38]. For concreteness the results of Fig. 5 are obtained for a profile deduced from a simple metric proposed in [37]:

$$\xi(x) = 1 + \frac{1}{3}\sin\left(\frac{4\pi x}{L}\right) + \frac{1}{3}\cos\left(\frac{2\pi x}{L}\right), \tag{69}$$

represented in Figs. 5(a) and (b). Note that such a profile is slowly varying, and does not yield abrupt changes of metric: we do not expect the type of anomalous corrections due to quantum fluctuations discussed in the previous section after a single quench. Yet, we will see that the succession of such quenches leads to manifestations of the gravitational anomaly.

For a period $t_p$ comparable with the time of flight for particles across the system, two distinct dynamical phases are reached at long-time depending on the relative magnitudes $t_2/t_1$ [35, 36]. A heating and non-heating phases are characterized by the evolution of the total energy $E_{\text{tot}} = L\bar{\varepsilon}$ of the closed system, which either grows exponentially or oscillates. Furthermore in the heating phase the energy density becomes highly inhomogeneous, localizing exponentially around a few spatial fixed points [35, 36].

First, following the discussion in Sec. 5.1, we realize that the periodic modulation of energy density of Eq. (68) can be realized by a periodic sequence of thermal quenches, with a profile $T(x)$ obtained by solving Eq. (58), provided this profile is always positive. Let us now notice that the Floquet drive Eq. (68) enforces a time-periodic quenches of a metric (5) with $f_2(x,t) = 1$ and

$$f_1(x,t) = \begin{cases} 1, & \text{for } t \in 0 < t < t_1, \\ 1/\xi^2(x), & \text{for } t \in t_1 < t < t_p. \end{cases} \tag{70}$$

Proceeding as in the single quench of the previous section, we solve the time-evolution of the energy momentum tensor stepwise, and apply suitable continuity equation. In doing so we access the energy and momentum density which are plotted in Fig 5(c) and (d), respectively, up to $t = 4t_p$. Three stroboscopic times are shown in Figs. 5(f) and (g). In these plots, the Hamiltonian $H_0$ was chosen as that of free Dirac fermions, with the duration of the two steps of metric chosen such that $Lt_1/v_F = 0.1$ and $Lt_2/v_F = 0.45$. Note that during step 2, the average velocity $\bar{v}_F$ is defined as $1/\bar{v}_F = \int_0^L dx/v(x) = \int_0^L dx\,\xi(x)/v_F$ such that the time of

flight across the circle of the particles is exactly one period: $L = v_F t_1 + \bar{v}_F t_2$, corresponding to the conditions to realise a heating phase [35, 37].

Focusing on the heating phase, we show that several of its features are manifestations of quantum fluctuations and can be traced back to gravitational anomalies. Indeed the gravitational anomaly contributes to the exponential growth of the average energy density. To show this we plot the total energy density $E_{\text{tot}} = \bar{\varepsilon} L$ at stroboscopic times in Fig. 5(e), extracted from Fig. 5(c). Plotted in Log scale, it shows a clear linear trend as a function of time. To highlight the contribution of the gravitational anomaly, in Fig. 5(e) we have separated two contributions: that arising from the classical Tolman-Ehrenfest temperature, and that directly linked to the gravitational anomaly. We observe that both have the same order of magnitude at large times.

A second signature of the gravitational anomaly is apparent in the spatial profile of the energy density, shown in Fig. 5(f) for stroboscopic times. The inset shows that the energy density can be locally negative, while satisfying that the total energy is always positive (Fig. 5(e)). Without quantum effects, the classical Tolman-Ehrenfest contribution $\varepsilon > 0$ for all $x$. This can be seen by noting that without the anomalous contribution $\varepsilon_q^{(1)}$ to (27a) the energy density is always positive for all $x$. However, in Fig. 5(f) we see that this is not the case, a clear manifestation of the gravitational and scale anomaly, reminiscent of the negative energy density close to the horizon of a black hole, as shown in Fig. 2.

We expect this relation between quantum properties of black holes and Floquet heating states to be generic. Indeed, the authors of Ref. [36] noted the relation between the effective metric of a sine squared Floquet CFT and that of two black holes at the accumulation points. This is in agreement with the manifestation of the trace and gravitational anomalies that we identified, and in particular with the negative density of energy close to these accumulation points, reminiscent of the black hole atmosphere.

# 6 Discussion

Let us start this discussion by commenting the conditions of application of our approach. Crucially, the notion of local temperature $T(x)$ requires some local energy relaxation, on scales smaller that the characteristic scales of variations of $T(x)$. In the context of the black hole, discussed in section 3, this local equilibration is assumed to occur locally, on a scale smaller that the curvature radius $\sqrt{R}$ from (18). While close to the black hole only the outgoing flux of Hawking's radiation need to be locally equilibrated, in the condensed matter examples of sections 4 and 5, we have assumed a single local temperature $T(x)$ common to left and right moving excitations, while still describing their motion as ballistic. This corresponds to a situation where the forward *inelastic* scattering occurs on scales much smaller that the backscattering between left and right movers, effectively neglected in this paper. This imposes a condition on the scattering potential, whose $2k_F$ components should be negligeable compared to the $q \simeq 0$ components. In more detail, denoting by $\ell_f$ and $\ell_b$ the forward and back scattering lengths, a sufficient condition for the excitations to be at a local thermal equilibrium amounts to consider a small enough thermal gradient satisfying $\ell_f \ll \tilde{l}_T \ll \ell_b$ in terms of the length $\tilde{l}_{T_{ext}}$ defined in (62). In practice, in a system with a fixed velocity $v_F$ temperature $T$ and scattering time $\tau$, and size L our theory will apply if the temperature difference between both end of the system satisfies $\frac{\Delta T}{\bar{T}} \ll \frac{L}{v_F \tau_{intra}}$. The situation where left and right movers are equilibrated at two different temperatures amounts to introduce a chiral temperature and thus a chiral metric, which goes beyond the scope of the present paper and will be presented in a forthcoming work. Probing experimentally the quench dynamics described in section 5 requires monitoring in time a local temperature which remains challenging. Such heat waves can be addressed within Pump-

Probe microscopy measurements, which consists in heating locally a material with a *e.g.* a laser pulse and unveiling the resulting heat dynamics by measuring the diffusion of a probe laser signal, see *e.g.* [89,90]). In the case of weak electron-phonon couplings, the dynamics of the electronic-excitation will be approximatively described by a quench procedure analogous to that of sec. 5.

In this work we have discussed that observable imprints on the thermal current and energy densities of anomalous quantum fluctuations at the origin of gravitational anomalies in field theory. These imprints manifest naturally in curved spacetime, such as the neighborhood of black-holes. However, extending Luttinger's correspondence beyond the realm of perturbative response theory, we have shown how they emerge, as naturally, in a flat spacetime when subjected to a single or periodic temperature quenches. The reason is that the equilibrium temperature profile in all the above situations can be phrased as an anomalous Tolman-Ehrenfest temperature, an equilibrium temperature profile that upgrades the classical result by Tolman and Ehrenfest by incorporating the quantum energy scales originating from gravitational anomalies. By using the anomalous Tolman-Ehrenfest temperature we were able to derive a modified Luttinger relation equivalence between strongly curved spacetime and large temperature variations. Within the realm of response theory, the historical playground for the Luttinger equivalence, we showed that the relation between thermal and gravitational field gradients becomes non-linear when thermal gradients vary too strongly. This leads to new contributions to non-linear thermal conductivities.

It is important to stress that our results are not specific to 1+1 dimensional systems. The anomalous Tolman-Ehrenfest temperature can be defined in any dimension (see Appendix C). However, the case 1+1 dimensions is special, as gravitational anomalies alone specify the energy-momentum tensor. In higher dimensions the energy-momentum tensor is not sufficiently constrained by the gravitational anomalies. Additional requirements from *e.g.*, symmetries of the problem, have to be analyzed in a case to case basis [91] and deserve a separate study.

Beyond response theory, our work provides further insight on a variety of questions. For example, the precise connection between anomalies, the Luttinger relation and the magneto-thermal transport in Weyl semimetals was a matter of debate [10,11]. Specifically, it was so far unclear what was the relation between the Luttinger trick, and the gravitational anomaly contribution to thermal transport. Our work clarifies the situation by exemplifying how the $T^2$ contribution to the thermal current, induced by the gravitational anomaly, coincides with the anomalous Tolman-Ehrenfest temperature.

Additionally, our work shows how the role played by anomalous fluctuations induced by spacetime curvature was overlooked in a variety of physical situations. Notably, this includes periodic space-dependent shaping of interactions in closed 1D systems [35–38]. More generally, the mapping between space dependent quenching of the Hamiltonian parameters and thermal quenches that we identify and exploit in this work, serves as a systematic way to find new situations where gravitational anomalies play a role in flat-spacetime. In this work we have discussed a thermal quench and Floquet conformal field theories, but we expect quite generally that any system with sufficiently strongly varying temperature or parameter profiles will display properties of anomalous fluctuations similar to those inside a black hole's quantum atmosphere. This open new perspectives to experimentally test signatures of gravitational anomalies, and study Hawking radiation in a controlled environment.

Indeed, the largest Hawking temperature of the smallest astrophysical black holes are typically in the 50nK range: their anomalous thermal properties are difficult to detect. To observe black hole related phenomena, it is often appealing to resort to acoustic [69, 92–94], optical [95] or quantum fluid [96] black-hole analogues. Using of the extended Luttinger equivalence our work points to a new class of less obvious candidate systems. In the main

text we discussed Floquet thermal drives, whose timescales suggest they could be realizable in ultra-cold atomic experiments [36, 97] or in quantum wires heated by laser pulses.

# Acknowledgments

We thank S. Ciliberto, M. Geiler and E. Livine for insightful discussion, M. Vozmediano for useful comments, and J. Gooth and S. Galeski for sharing their experimental data, from which this project was born.

**Funding information**  A. G. G acknowledges financial support from the European Union Horizon 2020 research and innovation program under grant agreement No 829044 (SCHINES), and from the French National Research Agency under the grant ANR-18-CE30-0001-01 (TOPODRIVE). B.B. and D.C. acknowledges financial support from the IDEX Lyon Breakthrough program under the grant ToRe.

# A  momentum energy tensor and gravitational anomalies

In 1977, Christensen and Fulling [25] computed the stationary momentum energy tensor for both a 3+1 and a 1+1 dimensional Schwarzschild black hole. In this section, we show how their method allows to determine the stationary momentum energy tensor of a chiral field in any background 1+1 dimensional static metric. We highlight consequences of gravitational anomalies on these results.

## A.1  Gravitational anomalies

Focusing on a symmetric momentum energy tensor, the energy conservation of chiral fields, in the presence of a background metric, can be written (see also Eq. (19)) as

$$\nabla_\mu \mathcal{T}^{\mu\nu} = C_g \frac{\hbar v_F}{96\pi} \frac{1}{\sqrt{\left|\det\left(g_{\rho\sigma}\right)\right|}} \varepsilon^{\nu\mu} \nabla_\mu R, \tag{A.1}$$

where R represents the Ricci scalar curvature, $\varepsilon^{\mu\nu}$ is the totally antisymmetric tensor with $\varepsilon^{01} = 1$, and $C_g$ is the gravitational anomaly coefficient

$$C_g = \sum \chi c, \tag{A.2}$$

with $c$ the central charge and $\chi$ the chirality. In other words, this corresponds to $C_g = c_+ - c_-$, as defined in (16). This coefficient needs to be distinguished from the Weyl anomaly coefficient $C_w = \sum c = c_+ + c_-$ defined in (16). A simple proof of the formula (A.1) is derived in [21] for free chiral fermions with $C_g = +1$ and $C_g = -1$.

## A.2  Background metric properties

As mentioned in the main text, for convenience we will focus on the case of a diagonal, static metric given by

$$ds^2 = f_1(x)v_F^2 dt^2 - f_2(x)dx^2. \tag{A.3}$$

Even though in two dimension and for any 1+1 dimensional manifold, there exist global coordinates in which the metric is of the form

$$ds^2 = \Omega(x,t)^2 \left(v_F^2 dt^2 - dx^2\right), \tag{A.4}$$

we will stick to the diagonal metric (A.3), convenient to express the results in the original laboratory coordinates.

In the metric (A.3), the non zero Christoffel symbols

$$\left\{ \begin{matrix} \nu \\ \rho\mu \end{matrix} \right\} = \frac{1}{2} g^{\nu\sigma} \left( \partial_\rho g_{\sigma\mu} + \partial_\mu g_{\rho\sigma} - \partial_\rho g_{\rho\mu} \right), \tag{A.5}$$

are

$$\left\{ \begin{matrix} 0 \\ x0 \end{matrix} \right\} = \left\{ \begin{matrix} 0 \\ 0x \end{matrix} \right\} = \frac{1}{2} \frac{\partial_x f_1}{f_1},$$

$$\left\{ \begin{matrix} x \\ 00 \end{matrix} \right\} = \frac{1}{2} \frac{\partial_x f_1}{f_2}, \tag{A.6}$$

$$\left\{ \begin{matrix} x \\ xx \end{matrix} \right\} = \frac{1}{2} \frac{\partial_x f_2}{f_2}.$$

The corresponding non-zero Riemann tensor coefficients

$$R^\mu{}_{\nu\rho\sigma} = \partial_\rho \left\{ \begin{matrix} \mu \\ \nu\sigma \end{matrix} \right\} - \partial_\sigma \left\{ \begin{matrix} \mu \\ \nu\rho \end{matrix} \right\} + \left\{ \begin{matrix} \lambda \\ \nu\sigma \end{matrix} \right\} \left\{ \begin{matrix} \mu \\ \lambda\rho \end{matrix} \right\} - \left\{ \begin{matrix} \lambda \\ \nu\rho \end{matrix} \right\} \left\{ \begin{matrix} \mu \\ \lambda\sigma \end{matrix} \right\}, \tag{A.7}$$

are

$$R^0{}_{x0x} = -R^0{}_{xx0}$$
$$= -\frac{1}{2} \left( \frac{\partial_x^2 f_1}{f_1} - \frac{1}{2} \left( \frac{\partial_x f_1}{f_1} \right)^2 - \frac{1}{2} \frac{\partial_x f_1}{f_1} \frac{\partial_x f_2}{f_2} \right), \tag{A.8}$$

$$R^x{}_{0x0} = -R^x{}_{00x}$$
$$= \frac{1}{2} \left( \frac{\partial_x^2 f_1}{f_2} - \frac{1}{2} \frac{(\partial_x f_1)^2}{f_1 f_2} - \frac{1}{2} \frac{(\partial_x f_1)(\partial_x f_2)}{f_2^2} \right). \tag{A.9}$$

Hence the curvature Ricci scalar reads

$$R = g^{\rho\nu} R^\mu{}_{\nu\mu\rho}$$
$$= \frac{\partial_x^2 f_1}{f_1 f_2} - \frac{1}{2} \frac{\partial_x f_1}{f_1 f_2} \left[ \frac{\partial_x f_1}{f_1} + \frac{\partial_x f_2}{f_2} \right]. \tag{A.10}$$

### A.3 Momentum energy tensor

Let us now consider the conservation equation (A.1) in this curved spacetime. Considering the stationary solution ($\partial_0 T^\mu{}_\nu = 0$), we rewrite the two equations (for $\nu = 0, x$) in the coordinates (A.3) as

$$\partial_x \mathcal{T}^x{}_0 + \frac{1}{2} \frac{\partial_x f_2}{f_2} \mathcal{T}^x{}_0 - \frac{1}{2} \frac{\partial_x f_1}{f_2} \mathcal{T}^0{}_x = C_g \frac{\hbar v_F}{96\pi} \sqrt{\frac{f_1}{f_2}} \partial_x R, \tag{A.11}$$

$$\partial_x \mathcal{T}^x{}_x + \frac{1}{2} \frac{\partial_x f_1}{f_1} \mathcal{T}^x{}_x - \frac{1}{2} \frac{\partial_x f_1}{f_1} \mathcal{T}^0{}_0 = 0. \tag{A.12}$$

Using the symmetry properties $\mathcal{T}_{\mu\nu} = \mathcal{T}_{\nu\mu}$, expressed as

$$\mathcal{T}^0{}_x = -\frac{f_2}{f_1} \mathcal{T}^x{}_0, \tag{A.13}$$

and, the trace anomalies in 1+1 dimension, relating the trace of the energy momentum tensor to the spacetime geometry for conformal theories in 1+1 dimensions

$$\mathcal{T}^\alpha_{\ \alpha} = \mathcal{T}^0_{\ 0} + \mathcal{T}^x_{\ x} = C_w \frac{\hbar v_F}{48\pi} R \,, \tag{A.14}$$

we simplify (A.11, A.12) into

$$\partial_x \left[ \sqrt{f_1 f_2} \mathcal{T}^x_{\ 0} \right] = C_g \frac{\hbar v_F}{96\pi} f_1 \partial_x R \,, \tag{A.15a}$$

$$\partial_x \left[ f_1 \mathcal{T}^x_{\ x} \right] = C_w \frac{\hbar v_F}{96\pi} R \partial_x f_1 \,. \tag{A.15b}$$

These equations imply that the most general momentum energy tensor is

$$\mathcal{T}^\mu_{\ \nu} = \left( T^\mu_{\ \nu} \right)_0 + \left( T^\mu_{\ \nu} \right)_{an} \,, \tag{A.16}$$

$$\left( \mathcal{T}^\mu_{\ \nu} \right)_0 = \begin{pmatrix} \frac{C_0}{f_1} & \frac{C_1}{\sqrt{f_1 f_2}} \\ -\frac{f_2}{f_1} \frac{C_1}{\sqrt{f_1 f_2}} & -\frac{C_0}{f_1} \end{pmatrix} \,, \tag{A.17}$$

$$\left( \mathcal{T}^\mu_{\ \nu} \right)_{an} = \frac{\hbar v_F}{96\pi} \begin{pmatrix} C_w \left( 2R - \frac{1}{f_1} \int dx R \partial_x f_1 \right) & C_g \sqrt{\frac{f_1}{f_2}} \left( R - \frac{1}{f_1} \int dx R \partial_x f_1 \right) \\ -C_g \sqrt{\frac{f_2}{f_1}} \left( R - \frac{1}{f_1} \int dx R \partial_x f_1 \right) & C_w \frac{1}{f_1} \int dx R \partial_x f_1 \end{pmatrix} \,, \tag{A.18}$$

with $C_0$ and $C_1$ two constants to be fixed by boundary conditions.

# B  Dirac fermions in $d = 1 + 1$ curved spacetime

Although our considerations in the main text are applicable to any 1+1 dimensional conformal field theory, straightforward realizations of 1+1 dimensional fields appearing in the edge modes of 2+1 dimensional topological insulators or in Luttinger liquids are described by a free, massless Dirac Hamiltonian. Therefore, it is worth considering this case in more detail.

## B.1  Lagrangian

We consider a curved spacetime of metric

$$ds^2 = f_1(x) dt^2 - f_2(x) dx^2 \,. \tag{B.1}$$

The Lagrangian describing Dirac fermions in curved spacetime is given by

$$\mathcal{L}_g = \frac{i\hbar v_F}{2} e^\mu_a(x) \left( \bar{\psi} \gamma^a \overleftrightarrow{\partial}_\mu \psi \right) \,, \tag{B.2}$$

with the associated action

$$\mathcal{S}_g = \int dx^2 \det \left( e^a_\mu \right) \mathcal{L}_g \,, \tag{B.3}$$

where $\psi$ is a Dirac spinor, $\bar{\psi} = \psi^\dagger \gamma^0$, $e^\mu_a$ denotes the zweibein defined by $e^\mu_a g_{\mu\nu} e^\nu_b = \eta_{ab}$, $e^\mu_a e^b_\mu = \delta^b_a$, and $e^\mu_a e^a_\nu = \delta^\mu_\nu$. In this symmetrized version of the Lagrangian, the spinor connection

$$\omega^a_{\ b\mu} = e^a_\nu \nabla_\mu e^\nu_b = e^a_\nu \left( \partial_\mu e^\nu_b + \left\{ \begin{matrix} \nu \\ \rho\mu \end{matrix} \right\} e^\rho_b \right) \,, \tag{B.4}$$

does not appear.

An equivalent action (up to a boundary term) is obtained by an integration by parts:

$$\tilde{\mathcal{L}}_g = i\hbar v_F \bar{\psi} \gamma^a \left( e_a^\mu(x) \partial_\mu + \frac{1}{2\det(e)} \partial_\mu \left[ e_a^\mu \det(e) \right] \right) \psi \,, \tag{B.5}$$

restoring the dependence in the spinor connection since

$$\tilde{\mathcal{L}}_g = i\hbar v_F \bar{\psi} \gamma^a e_a^\mu(x) \left( \partial_\mu + \Omega_\mu \right) \psi \,, \tag{B.6}$$

$$\Omega_\mu = \frac{1}{8} \omega_{ab\mu} \left[ \gamma^a, \gamma^b \right] \,, \tag{B.7}$$

where in 1+1 dimensions

$$\gamma^a e_a^\mu \Omega_\mu = \frac{1}{2\det(e)} \partial_\mu \left[ e_a^\mu \det(e) \right] \,. \tag{B.8}$$

In the case of the metric (B.1), the above action simplifies into

$$\mathcal{S}_g = \frac{i\hbar v_F}{2} \int \mathrm{d}x^2 \left[ \frac{\sqrt{f_2}}{v_F} \psi^\dagger \overleftrightarrow{\partial}_t \psi + \sqrt{f_1} \left( \psi^\dagger \gamma^0 \gamma^x \overleftrightarrow{\partial}_x \psi \right) \right] \,. \tag{B.9}$$

## B.2 Hamiltonian

The conjugate momentum associated to $\psi$ and $\psi^\dagger$ is defined, respectively, by

$$\pi^\dagger = \frac{\delta S}{\delta \partial_0 \psi} = \frac{i\hbar v_F}{2} \det(e) \psi^\dagger \gamma^0 \gamma^a e_a^0 \,, \tag{B.10}$$

$$\pi = \frac{\delta S}{\delta \partial_0 \psi^\dagger} = -\frac{i\hbar v_F}{2} \det(e) \gamma^0 \gamma^a e_a^0 \psi \,. \tag{B.11}$$

The Hamiltonian density is therefore defined by

$$\begin{aligned} \mathcal{H}(x) &= \pi^\dagger \partial_0 \psi + \partial_0 \psi^\dagger \pi - \det(e) \mathcal{L} \\ &= -\det(e) \frac{i\hbar v_F}{2} e_a^x \left( \bar{\psi} \gamma^a \overleftrightarrow{\partial}_x \psi \right) \,. \end{aligned} \tag{B.12}$$

For the above metric, we thus get a momentum operator

$$\pi^\dagger = \frac{i\hbar v_F}{2} \sqrt{f_2} \psi^\dagger \,, \qquad \pi = -\frac{i\hbar v_F}{2} \sqrt{f_2} \psi \,, \tag{B.13}$$

and a Hamiltonian density

$$\mathcal{H}(x) = -\sqrt{f_1} \frac{i\hbar v_F}{2} \left( \psi^\dagger \gamma^0 \gamma^x \overleftrightarrow{\partial}_x \psi \right) \,. \tag{B.14}$$

## B.3 Scalar product and density

In a curved spacetime, the scalar product between two spinors $\phi$ and $\psi$ is defined through

$$\langle \phi | \psi \rangle = \int \mathrm{d}x \, \det \left( e_\mu^a \right) \bar{\phi} \, e_a^0 \gamma^a \psi \,, \tag{B.15}$$

where $e_a^0 \gamma^a$ is the curved spacetime matrix $\gamma^{\mu=0}$. From the total number of particles expressed as the scalar product $\langle \psi | \psi \rangle$, we deduce the particle density at position $x$, $n(x) = \det \left( e_\mu^a \right) \psi^\dagger \gamma^0 e_a^0 \gamma^a \psi$, which reads for our metric

$$n(x) = \sqrt{f_2} \, \psi^\dagger \psi \,. \tag{B.16}$$

## B.4 momentum energy tensor

### B.4.1 Symmetrized version

The momentum energy tensor is defined as

$$
\mathcal{T}_{\mu\nu} = \frac{1}{2\det(e)} \left( \frac{\delta S}{\delta e^{\mu}_a} e_{\nu a} + \mu \leftrightarrow \nu \right). \tag{B.17}
$$

From the Dirac action (B.3), we obtain

$$
\mathcal{T}_{\mu\nu} = \frac{i\hbar v_F}{4} \left[ e_{\nu a}(x) \left( \bar{\psi} \gamma^a \overleftrightarrow{\partial}_{\mu} \psi \right) + \mu \leftrightarrow \nu \right] - g_{\mu\nu} \left[ \frac{i\hbar v_F}{2} e^{\rho}_b(x) \left( \bar{\psi} \gamma^b \overleftrightarrow{\partial}_{\rho} \psi \right) \right]. \tag{B.18}
$$

Four our specific metric (B.1), this reduces to

$$
\det(e)\mathcal{T}^0_{\ 0} = -\sqrt{f_1} \frac{i\hbar v_F}{2} \left( \psi^{\dagger} \gamma^0 \gamma^x \overleftrightarrow{\partial}_x \psi \right) \equiv \mathcal{H}(x), \tag{B.19}
$$

$$
\begin{aligned}
\det(e)\mathcal{T}^{0x} &= \det(e)\mathcal{T}^{x0} \\
&= \frac{i\hbar v_F}{4} \left[ \frac{1}{v_F \sqrt{f_1}} \left( \psi^{\dagger} \gamma^0 \gamma^x \overleftrightarrow{\partial}_t \psi \right) - \frac{1}{\sqrt{f_2}} \left( \psi^{\dagger} \overleftrightarrow{\partial}_x \psi \right) \right],
\end{aligned} \tag{B.20}
$$

$$
\det(e)\mathcal{T}^1_{\ 1} = -\sqrt{f_2} \frac{i\hbar}{2} \left( \psi^{\dagger} \overleftrightarrow{\partial}_t \psi \right). \tag{B.21}
$$

### B.4.2 Non-symmetric version

A non-symmetric version of this tensor, obtained by varying the action with respect to the tetrad while keeping the spinor connection fixed, is defined as

$$
\begin{aligned}
\mathcal{T}^{\mu}_{\ a} &= -\frac{1}{\det(e)} \frac{\delta S}{\delta e^a_{\mu}} \\
&= \frac{i\hbar v_F}{2} e^{\rho}_a e^{\mu}_b \left[ \bar{\psi} \gamma^b \overleftrightarrow{\partial}_{\rho} \psi \right] - e^{\mu}_a \left[ \frac{i\hbar v_F}{2} e^{\rho}_b(x) \left( \bar{\psi} \gamma^b \overleftrightarrow{\partial}_{\rho} \psi \right) \right].
\end{aligned} \tag{B.22}
$$

Alternatively, we can use the related quantities

$$
\mathcal{T}^a_{\ \mu} = \frac{1}{\det(e)} \frac{\delta S}{\delta e^{\mu}_{\ a}} \equiv e^a_{\rho} \mathcal{T}^{\rho}_{\ b} e^b_{\mu}. \tag{B.23}
$$

When working in curved space, it is often useful to express the non-symmetric version of the momentum energy tensor with only curved spacetime indices, corresponding to

$$
\tilde{\mathcal{T}}^{\mu}_{\ \nu} = \mathcal{T}^{\mu}_{\ a} e^a_{\nu}, \tag{B.24}
$$

from which we deduce the expression of the energy, the density of current of energy, etc. Note that the symmetrized version (B.17) can be recovered as

$$
\mathcal{T}_{\mu\nu} = \tilde{\mathcal{T}}_{\mu\nu} + \tilde{\mathcal{T}}_{\nu\mu}. \tag{B.25}
$$

For our metric (B.1) we obtain:

$$
\det(e)\tilde{\mathcal{T}}^0_{\ 0} = -\sqrt{f_1} \frac{i\hbar v_F}{2} \left( \psi^{\dagger} \gamma^0 \gamma^x \overleftrightarrow{\partial}_x \psi \right), \tag{B.26}
$$

$$
\det(e)\tilde{\mathcal{T}}^1_{\ 0} = \sqrt{f_1} \frac{i\hbar v_F}{2} \left( \psi^{\dagger} \gamma^0 \gamma^x \overleftrightarrow{\partial}_0 \psi \right), \tag{B.27}
$$

$$\det(e)\tilde{\mathcal{T}}^0_{\ 1} = \sqrt{f_2}\frac{i\hbar v_F}{2}\left(\psi^\dagger\overleftrightarrow{\partial}_x\psi\right),\tag{B.28}$$

$$\det(e)\tilde{\mathcal{T}}^1_{\ 1} = -\sqrt{f_2}\frac{i\hbar v_F}{2}\left(\psi^\dagger\overleftrightarrow{\partial}_0\psi\right).\tag{B.29}$$

While this tensor may no longer look symmetric, its averages on-shell are indeed symmetric (*i.e.* for fields satisfying the equation of motion).

The operators for the density energy, momentum, energy current and pressure are therefore

$$\varepsilon = \tilde{\mathcal{T}}^0_{\ 0} = -\frac{1}{\sqrt{f_2}}\frac{i\hbar v_F}{2}\left(\psi^\dagger\gamma^0\gamma^x\overleftrightarrow{\partial}_x\psi\right),\tag{B.30}$$

$$J_\varepsilon = v_F\det(e)\tilde{\mathcal{T}}^{10} = \frac{1}{\sqrt{f_1}}\frac{i\hbar v_F^2}{2}\left(\psi^\dagger\gamma^0\gamma^x\overleftrightarrow{\partial}_0\psi\right),\tag{B.31}$$

$$\Pi = \frac{1}{v_F}\det(e)\tilde{\mathcal{T}}^{01} = -\frac{1}{\sqrt{f_2}}\frac{i\hbar}{2}\left(\psi^\dagger\overleftrightarrow{\partial}_x\psi\right),\tag{B.32}$$

$$p = -\tilde{\mathcal{T}}^1_{\ 1} = \frac{1}{\sqrt{f_1}}\frac{i\hbar v_F}{2}\left(\psi^\dagger\overleftrightarrow{\partial}_0\psi\right).\tag{B.33}$$

# C  Trace anomaly and thermodynamics in d+1 dimensions

In this appendix, we consider the consequences of a non-vanishing energy momentum trace on the thermodynamics of an isotropic medium with a single radiative pressure $p$.

From the thermodynamic relation $dE = TdS - pdV$, we deduce the relation between densities

$$\varepsilon = \left.\frac{dE}{dV}\right|_T = T\left.\frac{dS}{dV}\right|_V - p \;\Rightarrow\; \varepsilon + p = T\left.\frac{dS}{dV}\right|_T = T\left.\frac{dp}{dT}\right|_V.\tag{C.1}$$

In d+1 dimensions, the trace of the momentum energy tensor $\mathcal{T}^\mu_{\ \mu}$ is expressed in terms of the energy density $\varepsilon$ and pressure $p$ as $\varepsilon = d\,p + \mathcal{T}^\mu_{\ \mu}$. In $1+1$ dimension, this trace is defined in (17) in terms of an energy density $\varepsilon^{(1)}_q$ of (24) as $\mathcal{T}^\mu_{\ \mu} = C_w\varepsilon^{(1)}_q$. Combining this relation with (C.1), we get

$$(d+1)p + \mathcal{T}^\mu_{\ \mu} = T\left.\frac{dp}{dT}\right|_V.\tag{C.2}$$

Given that $\mathcal{T}^\mu_{\ \mu}$ is independent on temperature, by integration of the above equation, we get

$$p = \lambda\,T^{d+1} - \frac{1}{d+1}\mathcal{T}^\mu_{\ \mu},\tag{C.3}$$

$$\varepsilon = \lambda d\,T^{d+1} + \frac{1}{d+1}\mathcal{T}^\mu_{\ \mu}.\tag{C.4}$$

These relations identify with the equations (27) for $d = 1$.

Let us now consider the entropy of the system, defined as $S = (E+F)/(TV) = (\varepsilon+p)/T$. From Eqs. (C.3,C.4) we get

$$S = \lambda(d+1)T^d\tag{C.5}$$

$$= \lambda(d+1)\left(\frac{\varepsilon}{\lambda d} - \frac{1}{\lambda d(d+1)}\mathcal{T}^\mu_{\ \mu}\right)^{\frac{d}{d+1}}.\tag{C.6}$$

Temperature and entropy are related through the relation $T^{-1} = dS/d\varepsilon$. Indeed, we check that the temperature entering the relations (C.3,C.4) satisfy this equality:

$$\frac{dS}{d\varepsilon} = \left( \frac{\varepsilon}{\lambda d} - \frac{1}{\lambda d(d+1)} \mathcal{T}^\mu_{\ \mu} \right)^{-\frac{1}{d+1}} = \frac{1}{T}. \tag{C.7}$$

Specifying these relations to $d = 1$, from the equations (27a,27b), the entropy reads $S = (\varepsilon + p)/T = 2\gamma T = 2\sqrt{\gamma(\varepsilon - \varepsilon_q^{(1)})}$. From this, we check that $dS/d\varepsilon = \sqrt{\gamma/(\varepsilon - \varepsilon_q^{(1)})} = 1/T$. This shows that whenever a temperature can be defined through the relations (C.3,C.4), it can be associated to a standard thermodynamics with a positive entropy. In the present paper, from the relation (28) it corresponds to the condition $\gamma T_{\text{TE}}^2 > -\varepsilon_q^{(2)}$. The fate of a system escaping this condition goes beyond the scope of the present paper.

# D Explicit solution of the anomalous Luttinger equivalence

In this appendix, we show how to identify a generic solution of eq. (32).

We consider the bulk of a thermal conductor, away from boundaries, where we assume the local temperature to vary as $T(x) = T_0(1 + a\ \tau(x))$. In a region of size $L$, setting $\tau(\pm L/2) = \pm 1/2$, we get $T_0 = (T_L + T_R)/2, a = 2(T_R - T_L)/(T_R + T_L)$ where $T_L = T(-L/2), T_R = T(+L/2)$. We will derive the energy current perturbatively in the parameter $a$. First, this amounts to identify the gravitational potential $\phi$ equivalent to this temperature profile, which satisfy

$$\frac{T^2(x)}{T_0^2} = e^{-2\phi(x)} + \lambda_{T_0}^2 \partial_x^2 \phi(x), \tag{D.1}$$

with a thermal lengthscale $\lambda_{T_0} = \frac{\hbar v_F}{2\pi k_B T_0}$.
A generic solution $\phi(x) = \sum_{n=0}^\infty a^n \phi^{(n)}(x)$ of eq. (D.1) satisfies

$$\phi^{(n)}(x) = a_n \sinh\left(\sqrt{2}\frac{x}{\lambda_{T_0}}\right) + b_n \cosh\left(\sqrt{2}\frac{x}{\lambda_{T_0}}\right) + \frac{1}{\sqrt{2}\lambda_{T_0}} \int_0^x \sinh\left(\sqrt{2}\frac{x-x'}{\lambda_{T_0}}\right) \alpha^{(n)}(x')dx'. \tag{D.2}$$

With $\alpha^{(n)}$ a source term which is set by $\tau(x)$ and the lower orders $\phi^{(m)}$ with $m < n$.

Let us focus on linear temperature profile, for which $\tau(x) = x/L$, such as $a \equiv L\partial_x T/T_0$. From (D.2) we get

$$\phi^{(1)}(x) = \left(a_1 + \frac{\lambda_{T_0}}{\sqrt{2}L}\right) \sinh\left(\sqrt{2}\frac{x}{\lambda_{T_0}}\right) + b_1 \cosh\left(\sqrt{2}\frac{x}{\lambda_{T_0}}\right) - \frac{x}{L}. \tag{D.3}$$

Imposing a finite gravitational potential in the thermodynamic limit $L \gg \lambda_{T_0}$ sets $a_1 = -\lambda_{T_0}/(\sqrt{2}L)$ and $b_1 = 0$, leading to $\phi^{(1)}(x) = -x/L$. Recursively, we get

$$\phi^{(1)}(x) = -\frac{x}{L}, \tag{D.4}$$

$$\phi^{(2)}(x) = \frac{1}{2L^2}\left(x^2 + \lambda_{T_0}^2\right), \tag{D.5}$$

$$\phi^{(3)}(x) = \frac{1}{L^3}\left(-\frac{1}{3}x^3 - 2x\lambda_{T_0}^2\right), \tag{D.6}$$

$$\phi^{(4)}(x) = \frac{1}{4L^4}\left(x^4 + 20x^2\lambda_{T_0}^2 + 21\lambda_{T_0}^4\right), \tag{D.7}$$

which corresponds to $\phi(x) = \phi_{\text{Lutt}} + \delta\phi$ where $\phi_{\text{Lutt}} = -\ln(T_0/T(x))$ is the potential deduced from the standard Luttinger equivalence, and $\delta\phi$ encodes the modifications of this equivalent potential induced by the gravitational anomalies, corresponding to the last term on eq. (D.1):

$$\delta\phi = \frac{a^2}{2L^2}\lambda_{T_0}^2 - 2\frac{a^3}{L^2}x\lambda_{T_0}^2 + \frac{a^4}{4L^4}\left[20x^2\lambda_{T_0}^2 + 21\lambda_{T_0}^4\right] + \mathcal{O}(a^5). \tag{D.8}$$

# E Gravitational anomaly and linear response

## E.1 Momentum operator in a static gravitational potential

Here we compute the current in a system at a reference temperature $T_0 > 0$, in the curved space described by the Luttinger metric, for a simple Hamiltonian density in real space

$$\hat{h} = -i\hbar v_F \frac{\overleftarrow{\partial_x} + \overrightarrow{\partial_x}}{2}. \tag{E.1}$$

The $D = 1 + 1$ Dirac Hamiltonian in curved space is given by

$$\mathcal{H} = \int \mathrm{d}x \, e^{\phi(x)} h(x), \tag{E.2}$$

where

$$h(x) = -i\hbar v_F \, \Psi^\dagger(x) \frac{\overleftarrow{\partial_x} + \overrightarrow{\partial_x}}{2} \Psi(x). \tag{E.3}$$

The expression of the momentum operator is then deduced from Eq. (B.32) with $f_2 = 1$, $\hat{\Pi} = \hat{h}/v_F$, expressed in Fourier components as:

$$\hat{\Pi}(x) = \int \frac{\mathrm{d}k\mathrm{d}q}{(2\pi)^2} \, e^{iqx} \Psi^\dagger_{k-\frac{q}{2}} \hat{\Pi}_{k-\frac{q}{2},k+\frac{q}{2}} \Psi_{k+\frac{q}{2}}, \tag{E.4}$$

with,

$$\hat{\Pi}_{k-\frac{q}{2},k+\frac{q}{2}} = \hbar k. \tag{E.5}$$

## E.2 Average Momentum

The average momentum at zero temperature can be computed equation from eqs. (E.4) and (E.5) as

$$\Pi(x) = \langle\hat{\Pi}(x)\rangle = -i\int \frac{\mathrm{d}k\mathrm{d}q}{(2\pi)^2}\frac{\mathrm{d}\omega}{2\pi} e^{iqx} \, \hat{\Pi}_{k-\frac{q}{2},k+\frac{q}{2}} G^<_{k+\frac{q}{2},k-\frac{q}{2}}(\omega), \tag{E.6}$$

where the lesser Green's functions $g^<$ and $G^<$ are defined (for $t > 0$) by

$$g^<_{k-q,k+q}(t) = i\Theta(t)\left\langle\Psi^\dagger_{k-q}(0)\Psi_{k+q}(t)\right\rangle, \tag{E.7}$$

and

$$G^<_{k-q,k+q}(\omega) = \int \mathrm{d}t \, e^{i\omega t} g^<_{k-q,k+q}(t). \tag{E.8}$$

This lesser Green's function is related to the retarded one, defined as

$$g^R_{p,q}(t) = -i\Theta(t)\left\langle\left\{\Psi^\dagger_q(0),\Psi_p(t)\right\}\right\rangle, \tag{E.9}$$

through the relation

$$G_{p,q}^{<}(\omega) = -2if(\omega)\,\text{Im}\left(G_{p,q}^{R}(\omega)\right), \tag{E.10}$$

with

$$f(\omega) = \frac{1}{1 + e^{\frac{\omega}{k_B T}}}. \tag{E.11}$$

Working in perturbation theory at first order in the gravitational potential $\phi(x)$, we develop the Green's function using the Dyson equation

$$G_{k',k}^{<}(\omega) = \left(G_0^{<}\right)_{k',k} + \int dx\, e^{i(k-k')x}\phi(x)\left(\left(G_0^{R}\right)_{k',k'}\hat{h}_{k',k}\left(G_0^{<}\right)_{k,k} + \left(G_0^{<}\right)_{k',k'}\hat{h}_{k',k}\left(G_0^{A}\right)_{k,k}\right), \tag{E.12}$$

where the dependence on the fixed frequency index $\omega$ has been omitted for clarity in the r.h.s. of the equation, and with $(G_0)_{k,k'}(\omega)$ the Green function in absence of perturbation

$$(G_0^{R/A})_{k,k'}(\omega) = \delta_{k,k'}[\omega - \hat{h}_{k,k} \pm i0^+]^{-1}, \tag{E.13}$$

$$(G_0^{<})_{k,k'}(\omega) = 2i\pi\delta_{k,k'}f(\omega)\delta(\omega - h_{k,k}). \tag{E.14}$$

By using the Dyson expansion of the Green's function (E.12) into the expression (E.6) we obtain the perturbative expansion in $\phi$ of the energy current. At temperatures $k_B T_0$ small compared to the energy range over which the system is well described by the Dirac linear Hamiltonian, we can develop the Fermi-Dirac distribution as

$$f(\omega) = \Theta(-\omega) - \frac{\pi^2}{6}k_B^2 T_0^2\,\partial_\omega\delta(\omega) + \mathcal{O}\left((k_B T_0)^4\right). \tag{E.15}$$

The equilibrium energy current density, corresponding to the the $0^{th}$ order term, can therefore be written as

$$\begin{aligned}
\Pi^{(0)}(x) &= -i\int \frac{dk}{2\pi}\frac{d\omega}{2\pi}\,\hat{\Pi}_{k,k}f(\omega)\left[\frac{1}{\omega - \hat{h}_{k,k} - i0^+} - \frac{1}{\omega - \hat{h}_{k,k} + i0^+}\right] \\
&= \overline{\Pi} + \frac{\text{sign}(v_F)}{v_F^2}\frac{\pi}{12\hbar}k_B^2 T_0^2 + O\left((k_B T_0)^4\right),
\end{aligned} \tag{E.16}$$

where $\overline{\Pi}$ is the Fermi sea contribution to the momentum density. The first order in $\phi$ contribution to the momentum density is given by

$$\begin{aligned}
\Pi^{(1)}(x) &= -i\int \frac{dk\,dq}{(2\pi)^2}\frac{d\omega}{2\pi}dy\,\hat{\Pi}_{k-\frac{q}{2},k+\frac{q}{2}}e^{iq(x-y)}\phi(y)\left\{\left(G_0^{R}\right)_{k+\frac{q}{2},k+\frac{q}{2}}\hat{h}_{k+\frac{q}{2},k-\frac{q}{2}}\left(G_0^{<}\right)_{k-\frac{q}{2},k-\frac{q}{2}}\right. \\
&\qquad\qquad\qquad\qquad\qquad\left. + \left(G_0^{<}\right)_{k+\frac{q}{2},k+\frac{q}{2}}\hat{h}_{k+\frac{q}{2},k-\frac{q}{2}}\left(G_0^{A}\right)_{k-\frac{q}{2},k-\frac{q}{2}}\right\} \\
&= -2\int \frac{dk\,dq}{(2\pi)^2}\frac{d\omega}{2\pi}dy\,\phi(y)f(\omega)\,\text{Im}\left[e^{iq(x-y)}\frac{\hat{\Pi}_{k-\frac{q}{2},k+\frac{q}{2}}\hat{h}_{k+\frac{q}{2},k-\frac{q}{2}}}{\left(\omega - \hat{h}_{k+\frac{q}{2},k+\frac{q}{2}} + i0^+\right)\left(\omega - \hat{h}_{k-\frac{q}{2},k-\frac{q}{2}} + i0^+\right)}\right].
\end{aligned} \tag{E.17}$$

The long range physics dominating the linear response theory is given by the first order in the development of

$$\begin{aligned}
\frac{\hat{\Pi}_{k-\frac{q}{2},k+\frac{q}{2}}\hat{h}_{k+\frac{q}{2},k-\frac{q}{2}}}{\left(\omega - \hat{h}_{k+\frac{q}{2},k+\frac{q}{2}} + i0^+\right)\left(\omega - \hat{h}_{k-\frac{q}{2},k-\frac{q}{2}} + i0^+\right)} &= \frac{v_F(\hbar k)^2}{(\omega - v_F\hbar k + i0^+)^2} \\
&\quad + \frac{v_F(\hbar k)^2}{(\omega - v_F\hbar k + i0^+)^4}\frac{(v_F\hbar q)^2}{4} + \mathcal{O}\left((\hbar q)^4\right).
\end{aligned} \tag{E.18}$$

The gradient expansion of the regularized current can therefore be written, after integration by parts on the variable $y$,

$$
\begin{aligned}
\Pi^{(1)}(x) &\approx -2 \int \frac{\mathrm{d}k\mathrm{d}q}{(2\pi)^2} \frac{\mathrm{d}\omega}{2\pi} \mathrm{d}y \, \phi(y) f(\omega) \\
&\quad \times \operatorname{Im}\left[ \frac{v_F (\hbar k)^2}{(w - v_F \hbar k + i0^+)^2} e^{iq(x-y)} - \frac{\hbar^2 v_F^2}{4} \frac{v_F (\hbar k)^2}{(w - v_F \hbar k + i0^+)^4} \partial_y^2 \left( e^{iq(x-y)} \right) \right] \\
&\approx -2 \int \frac{\mathrm{d}k\mathrm{d}q}{(2\pi)^2} \frac{\mathrm{d}\omega}{2\pi} \mathrm{d}y f(\omega) \\
&\quad \times \operatorname{Im}\left[ e^{iq(x-y)} \left\{ \frac{v_F (\hbar k)^2}{(w - v_F \hbar k + i0^+)^2} \phi(y) - \frac{\hbar^2 v_F^2}{4} \frac{v_F (\hbar k)^2}{(w - v_F \hbar k + i0^+)^4} \partial_y^2 (\phi(y)) \right\} \right] \\
&\approx -2 \int \frac{\mathrm{d}k}{2\pi} \frac{\mathrm{d}\omega}{2\pi} f(\omega) \operatorname{Im}\left[ \frac{v_F (\hbar k)^2}{(w - v_F \hbar k + i0^+)^2} \phi(x) - \frac{\hbar^2 v_F^2}{4} \frac{v_F (\hbar k)^2}{(w - v_F \hbar k + i0^+)^4} \partial_x^2 (\phi(x)) \right].
\end{aligned}
$$

By using

$$
\operatorname{Im}\left( \frac{(-1)^n n!}{(\omega - x \pm i0^+)^{n+1}} \right) = \mp \pi \partial_\omega^n [\delta(\omega - x)], \tag{E.19}
$$

we can express this momentum density as

$$
\begin{aligned}
\Pi^{(1)}(x) &\approx -\int \frac{\mathrm{d}k}{2\pi} \mathrm{d}\omega \left\{ \Theta(-\omega) - \frac{\pi^2}{6} k_B^2 T_0^2 \partial_\omega \delta(\omega) \right\} \Bigg[ v_F (\hbar k)^2 \partial_\omega \delta(\omega - h_{k,k}) \phi(x) \\
&\qquad\qquad\qquad\qquad\qquad\qquad - \frac{\hbar^2}{24} \partial_x^2 \phi(x) v_F^3 (\hbar k)^2 \partial_\omega^3 \delta(\omega - h_{k,k}) \Bigg] \\
&\approx -\int \frac{\mathrm{d}k}{2\pi} \mathrm{d}\omega \, \delta(\omega) \left[ v_F (\hbar k)^2 \delta(\omega - h_{k,k}) \phi(x) - \frac{\hbar^2 v_F^2}{24} \partial_x^2 \phi(x) \, v_F (\hbar k)^2 \partial_\omega^2 \delta(\omega - h_{k,k}) \right] \\
&\quad - \frac{\pi^2}{6} k_B^2 T_0^2 \int \frac{\mathrm{d}k}{2\pi} \mathrm{d}\omega \, \delta(\omega) \Bigg[ v_F (\hbar k)^2 \partial_\omega^2 \delta(\omega - h_{k,k}) \phi(x) \\
&\qquad\qquad\qquad\qquad\qquad\qquad - \frac{\hbar^2 v_F^2}{24} \partial_x^2 \phi(x) \, v_F (\hbar k)^2 \partial_\omega^4 \delta(\omega - h_{k,k}) \Bigg]. \tag{E.20}
\end{aligned}
$$

Using the replacement $\partial_\omega \delta(\omega - h_{k,k}) = -\frac{1}{\hbar v_F} \partial_k \delta(\omega - h_{k,k})$, we can integrate on $\omega$ to get

$$
\begin{aligned}
\Pi^{(1)}(x) &\approx -\int \frac{\mathrm{d}k}{2\pi} \left[ v_F (\hbar k)^2 \delta(h_{k,k}) \phi(x) - \frac{\hbar^2 v_F^2}{24} \partial_x^2 \phi(x) v_F (\hbar k)^2 \left( \frac{-1}{\hbar v_F} \right)^2 \partial_k^2 \delta(h_{k,k}) \right] \\
&\quad - \frac{\pi^2}{6} k_B^2 T_0^2 \int \frac{\mathrm{d}k}{2\pi} \Bigg[ v_F (\hbar k)^2 \left( \frac{-1}{\hbar v_F} \right)^2 \partial_k^2 \delta(h_{k,k}) \phi(x) \\
&\qquad\qquad\qquad\qquad\qquad\qquad - \frac{\hbar^2 v_F^2}{24} \partial_x^2 \phi(x) \, v_F (\hbar k)^2 \left( \frac{-1}{\hbar v_F} \right)^4 \partial_k^4 \delta(h_{k,k}) \Bigg] \\
&\approx -\int \frac{\mathrm{d}k}{2\pi} \left[ v_F (\hbar k)^2 \phi(x) - \frac{1}{12} \partial_x^2 \phi(x) v_F \hbar^2 \right] \delta(h_{k,k}) - \frac{\pi^2}{6 v_F^2} k_B^2 T_0^2 \int \frac{\mathrm{d}k}{2\pi} 2 v_F \phi(x) \delta(h_{k,k}) \\
&\approx \operatorname{sign}(v_F) \left\{ \frac{\hbar}{24\pi} \partial_x^2 \phi(x) - \frac{\pi}{6\hbar v_F^2} k_B^2 T_0^2 \phi(x) \right\}. \tag{E.21}
\end{aligned}
$$

Gathering the different terms, we obtain, at linear order in the gravitational field $\phi(x)$, that the momentum reads

$$
\Pi(x) = \bar{\Pi} + \operatorname{sign}(v_F) \left[ \frac{\pi}{12\hbar v_F^2} k_B^2 T_0^2 \left( 1 - 2\phi(x) \right) + \frac{\hbar}{24\pi} \partial_x^2 \phi(x) \right], \tag{E.22}
$$

which identifies, with the first order in $\phi$ of the full result (35b).

# F  Floquet quench dynamics and evolution of the momentum energy tensor

## F.1  Continuity equation during a metric quench

In this section, we study the continuity equation of the Lorentz symmetry breaking momentum energy tensor (21) during a quench of metric. For the sake of simplicity, and sticking to the protocols detailed in the main text, we consider a metric of the form

$$\mathrm{d}s^2 = f(x,t)v_F^2\mathrm{d}t^2 - \mathrm{d}x^2\,, \tag{F.1}$$

which is changed abruptly at $t = 0$:

$$f(x,t) = \begin{cases} f_I(x), & \text{for } t < 0, \\ f_{II}(x), & \text{for } t > 0\,. \end{cases} \tag{F.2}$$

Injecting these expression in the conservation equations (7a), together with the anomalies (17,21) leads to the conservation equations

$$\partial_0 \mathcal{T}^0{}_0 + \frac{1}{\sqrt{f}}\partial_x\left(\mathcal{T}^x{}_0\sqrt{f}\right) + \frac{\hbar v_F}{96\pi}\frac{C_g}{\sqrt{f}}R\partial_x f = 0\,,$$
$$\partial_0\left(\mathcal{T}^0{}_x\sqrt{f}\right) + \frac{1}{\sqrt{f}}\partial_x\left(\mathcal{T}^x{}_x f\right) - \frac{\hbar v_F}{96\pi}\frac{C_w}{\sqrt{f}}R\partial_x f = 0\,, \tag{F.3}$$

where we recall that $R(x) = \frac{\partial_x^2 f}{f} - \frac{1}{2}\left(\frac{\partial_x f}{f}\right)^2$. Integrating these equations between $t = 0^-$ and $t = 0^+$, we deduce that the variables which are continuous across the quench are both the energy density $\varepsilon = \mathcal{T}^0{}_0$ and the momentum density $\Pi_x = \frac{1}{v_F}\mathcal{T}^0{}_x\sqrt{f}$:

$$\mathcal{T}^0{}_0(0^-,x) = \mathcal{T}^0{}_0(0^+,x)\,, \tag{F.4}$$

$$\frac{1}{v_F}\sqrt{f_I(x)}\mathcal{T}^0{}_x(0^-,x) = \frac{1}{v_F}\sqrt{f_{II}(x)}\mathcal{T}^0{}_x(0^+,x)\,. \tag{F.5}$$

## F.2  Time evolution in a curved spacetime

The conservation equations (F.3) of $\varepsilon$ and $\Pi$, which are continuous at metric quenches, can be explicitly written using anomalies expressions (17,19) as:

$$\sqrt{f}\partial_0\varepsilon - v_F\partial_x(\Pi f) = \frac{\hbar v_F}{48\pi}C_g\left(f\partial_x R + \frac{1}{2}R\partial_x f\right) = \frac{\hbar v_F}{48\pi}C_g\partial_x\left(f\left(R - \bar{R}\right)\right)\,, \tag{F.6}$$

$$v_F\sqrt{f}\partial_0\Pi - \partial_x(\varepsilon f) = -\frac{\hbar v_F}{48\pi}C_w\partial_x\left(f\left(R - \bar{R}\right)\right)\,. \tag{F.7}$$

Defining $\mathcal{T}^{\pm} = \varepsilon \pm v_F\Pi$, we get

$$\sqrt{f}\partial_0\mathcal{T}^{\pm} \mp \partial_x\left[f\left(\mathcal{T}^{\pm} - \frac{\hbar v_F}{48\pi}c^{\pm}\left(R - \bar{R}\right)\right)\right] = 0\,, \tag{F.8}$$

where

$$\bar{R}(x) = \frac{1}{2f}\int_0^x R\partial_x f = \frac{1}{4}\left(\frac{\partial_x f}{f}\right)^2\,. \tag{F.9}$$

Hence, the evolution of $\varepsilon$ and $\Pi$ are deduced from two rules:

1. At the quenches, $\mathcal{T}^{\pm}$ is continuous.

2. Between quenches, since $f(x)$ does not depend on time, $\mathcal{T}^{\pm}$ satisfies the following equation of motion

$$\left(\partial_0 \mp \partial_y\right)\left[f\left(\mathcal{T}^{\pm} - \frac{\hbar v_F}{48\pi}c^{\pm}\left(R - \bar{R}\right)\right)\right] = 0 \,, \tag{F.10}$$

with a rescaled coordinate

$$y(x) = \int_0^x \frac{1}{\sqrt{f(u)}} du \,. \tag{F.11}$$

### F.3 Floquet stroboscopic evolution

We now derive the stroboscopic time evolution of $\mathcal{T}^{\pm}$ in a Floquet system from the previous equations of motion. We consider a metric (F.1) with

$$f(x, t) = \begin{cases} 1 & \text{for } t \in \,]nt_p, nt_p + t_1[ \,, \\ f(x) & \text{for } t \in \,]nt_p + t_1, (n+1)t_p[ \,. \end{cases} \tag{F.12}$$

where $t_p = t_1 + t_2$ and $n \in Z$. Calling $\mathcal{T}_n^{\pm}(x) = \mathcal{T}^{\pm}(x, nt_p)$ and applying the previous rules we get

$$\begin{aligned}\mathcal{T}^{\pm}(x, nt_p + t_1) &= \mathcal{T}^{\pm}(x \pm v_F t_1, nt_p) \\ &= \mathcal{T}_n^{\pm}(x \pm v_F t_1), \end{aligned} \tag{F.13}$$

and therefore,

$$\begin{aligned}\mathcal{T}_{n+1}^{\pm}(x) &\equiv \mathcal{T}^{\pm}(x, nt_p + t_1 + t_2) \\ &= \frac{f(x^{\pm})}{f(x)}\left[\mathcal{T}_n^{\pm}(x^{\pm} \pm v_F t_1) - \frac{\hbar v_F}{48\pi}c^{\pm}\left(R(x^{\pm}) - \bar{R}(x^{\pm})\right)\right] + \frac{\hbar v_F}{48\pi}c^{\pm}\left(R(x) - \bar{R}(x)\right) \,, \end{aligned} \tag{F.14}$$

with

$$x^{\pm}(x) = y^{-1}\left(y(x) \pm v_F t_2\right) \,. \tag{F.15}$$

We can rewrite the equation (F.14) in more compact form:

$$\mathcal{T}_{n+1}^{\pm}(x) = \left(\partial_x x^{\pm}\right)^2 \mathcal{T}_n^{\pm}\left(x^{\pm}\right) - \frac{\hbar v_F}{24\pi}c^{\pm}\left\{x^{\pm}, x\right\} \,, \tag{F.16}$$

where $\left\{x^{\pm}, x\right\}$ denotes the Schwarzian derivative of $x^{\pm}$ with respect to $x$,

$$\{f, x\} = \frac{\partial_x^3 f}{\partial_x f} - \frac{3}{2}\left(\frac{\partial_x^2 f}{\partial_x f}\right)^2 \,. \tag{F.17}$$

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
