# Peer review of "Anomalous Luttinger equivalence between temperature and curved spacetime: From black holes to thermal quenches"

_SciPost Physics, doi:SciPost Phys. 16, 084 (2024)_

## Round 1 · Referee Report · Anonymous (Referee 1) · 2023-1-25

Report

The manuscript discusses corrections to Luttinger relations due to quantum anomalies. It is clearly written, and the computations are easy to follow. In my opinion the only aspect of the manuscript that needs to be expanded is the discussion that the particles transport ballistically, and the two species don’t thermalize. This assumption for thermal systems seems to be very strong and in general valid for temperatures close to zero. Therefore, I would expect that it breaks down quite quickly as the temperature rises. However, I have not seen any discussion of the applicability of the formulas presented in the paper. Moreover, the validity of the corrections derived in this work requires that different species of fermions are kept at different temperatures. Again, no discussion of how close to realistic systems is this requirement, is included. I do not expect these relations to hold near black holes.

---

## Round 1 · Referee Report · Anonymous (Referee 2) · 2023-2-26

Report

This paper discusses nonlinear effects due to gravitational anomalies in 1+1d CFTs in a curved space, and the relationship with Luttinger's perspective on the relation between temperature gradients and gravity. My honest sense is that this paper is in some sense a commentary on well-established ideas about thermal CFTs in a curved background, and remarking on the breakdown of the calculation from Luttinger's linear response calculation? That could be valuable to the literature.

I think my main concern - or at least confusion - about this manuscript is the following. A notion of local temperature is probably only well-defined in curved spacetime with slowly varying curvature. Equilibrium is sharply defined: I can ask what is the profile of <T^mu nu> given a classical gravitational background, and the anomalous energy-momentum equation subject to boundary conditions. Assuming no quantum gravity, I can exactly solve those equations without ambiguity. But then, it is far from clear to me whether particular experimental probes of temperature correspond to what the authors think they do? Some possibly semantic questions about whether some particular definition of temperature need to be corrected at the nonlinear level, might not be that interesting to me personally, because I do not know whether the authors' definition of temperature is the one probed by any given experimental system. As one example, the discussion around Eq. 56 and 57 to me seems very possibly compatible with this idea: I start with some sharply varying temperature profile in flat space and, depending on interpretation, may assign it different notions of "local temperature"?

What an experiment could clearly (in principle) measure are the correlation functions of <T^mu nu>, which the authors calculate. But then, this calculation seems to be based on an assumption of how an external temperature profile would couple to the theory, and again, I am not sure how confident I am in that theory on arbitrarily short length scales. Maybe I am the one who does not understand.

I am confused around Eq. (53). Are the authors assuming that the temperature gradient is constant while the energy current is not? That seems to require a very special type of heat bath. It could just as easily be that the actual temperature varies in some nonlinear way, so that the energy current is better behaved. I do not see how this discussion can feasibly relate to most experimental systems, where if anything, it seems more plausible to argue some systems are approximately thermally isolated when electron-phonon coupling is weak. In this setting it is J_E that must be constant, while T(x) could vary quite nonlinearly. If the system is not thermally isolated, then the thermodynamics of the phonon bath must also be accounted for?

In Section 5.2, the authors have a discussion of dynamics after a quench. From my perspective this sounds like a very straightforward problem where gravity is not needed at all: there is a T_L and T_R associated to the left and right movers of the CFT, and the profiles of T_L and T_R simply propagate at velocity c. If this is all the authors are commenting on from the gravitational perspective, it would be helpful to understand the role that the anomaly is playing more carefully by adding some concrete equations showing the imprint of the anomaly after the spacetime becomes flat? For example, if I encode some initial inhomogeneity in energy density/energy current via coupling to background spacetime, and then quickly shut off this coupling and return to flat space, presumably once the coupling is shut off the anomalous terms disappear from the energy momentum equation. Then, is there any anomalous part to the energy-momentum tensor after the quench? I fail to see how it is possible, once the quench ends the spacetime would be flat? Perhaps again, the only way the anomaly is encoded is in the precise way the theory is coupled to the external heat bath, and it's far from clear to me that there is only one logical way to couple to this heat bath, especially if the proposed temperature varies on extremely short length scales.

I think this commentary should be addressed or refuted in a revised manuscript, but I would probably encourage publication afterwards.

---

## Round 2 · Referee Report · Anonymous (Referee 1) · 2023-11-14

Report

I recommend the paper for publication in the revised form.

---

## Round 2 · Referee Report · Anonymous (Referee 2) · 2023-12-14

Report

At some level, I don't know that I am fully sated by the authors' response, at the level of the physics. But in the spirit that the paper could be interesting and at leasts presents a *model* that is arguably falsifiable, I think it is fine to go ahead and publish, given the other positive report. Below I will state the points that I am still a bit unsure about. It might be useful for the authors to try and make simple changes to modify the manuscript in response to my suggestions below but I would not want to write another report just on those!

When the authors write e.g. Eq. (63), my expectation would be that there is a part of the current coming from the CFT and a part the authors attribute to quantum corrections. Are these terms parametrically larger than e.g. other corrections to CFT, in the condensed matter setting? I expect that the answer could be yes in certain regimes, it might be good to quantify what they might be (using numbers for the system that the authors quoted in another context in their reply).

I don't agree with the authors' claim that the physics is either ballistic or localized -- this does not sound correct for interacting systems. While often the 1+1D CFTs that arise in condensed matter are effectively free fermion-like, the authors make a point that their theory holds in an intermediate regime, where interactions cannot be neglected!

I also didn't quite follow the authors' comment about temperature in their response: if the system is held in contact with a bath, how do I know this bath doesn't thermalize the left and right movers together (which the authors assumed didn't happen in the comments on neglecting backscattering).
  • validity: good
  • significance: good
  • originality: high
  • clarity: good
  • formatting: excellent
  • grammar: good

Author:  Bermond Baptiste  on 2024-01-29  [id 4291]

(in reply to Report 2 on 2023-12-14)
Category:
remark
answer to question
reply to objection

We thank both Referees for providing us with detailed reports. Below, we address the concerns raised Referee 2 below. For the sake of convenience, we address the remarks of the Referee 2 in quoting the full texts of the reports. For the same reason, we took a liberty to divide the report in parts, and answer them part by part.

"When the authors write e.g. Eq. (63), my expectation would be that there is a part of the current coming from the CFT and a part the authors attribute to quantum corrections. Are these terms parametrically larger than e.g. other corrections to CFT, in the condensed matter setting? I expect that the answer could be yes in certain regimes, it might be good to quantify what they might be (using numbers for the system that the authors quoted in another context in their reply)."

-> In this paper, we do not consider condensed matter corrections to CFT. However, if other corrections, beyond the thermal contribution $\gamma T^2$ and the contributions of the gravitational anomaly were to appear in such a setup, we believe that it would be possible to evaluate the relative magnitude of the correction by comparing the corresponding length scale to the one characteristic of the problem: $\lambda_T$, $l_T$ and $\tilde{l}_T$. Similar strategies were used in our paper to evaluate the relevance of the gravitational anomaly correction in section 5.1.2.
More precisely, in section 5.1.2, the characteristic length scale are: $\lambda_{T_0}=1.2*10^{-5}m$ while the maxima of the space varying lengthscales $l_T$ and $\tilde{l}_T$ are $l_T^{max}\approx\tilde{l}_T^{max}=5.4*10^{-5}m$ for the panel (c) and (d) while $l_T^{max}\approx\tilde{l}_T^{max}=3.1*10^{-5}m$ for the panel (e) and (f).

"I don't agree with the authors' claim that the physics is either ballistic or localized -- this does not sound correct for interacting systems. While often the 1+1D CFTs that arise in condensed matter are effectively free fermion-like, the authors make a point that their theory holds in an intermediate regime, where interactions cannot be neglected!"

-> According to Luttinger's liquid theory, in a 1+1 dimensional conductor and in the presence of interactions, electrons are no longer the low energy degrees of freedom. Instead, one needs to study the physics of emergent collective density and spin modes. At the chemical potential, the band dispersion of these quasiparticles will be linear and define a modified Fermi velocity $v'_F$, such as the excitations in the system will behave as free particles but with a renormalized Fermi velocity $v'_F$. This is the essence of the Luttinger liquid approach. We refer to the motion of these collectives particles as "ballistic" transport in the main text, i.e. ballistic transport of a CFT.

"I also didn't quite follow the authors' comment about temperature in their response: if the system is held in contact with a bath, how do I know this bath doesn't thermalize the left and right movers together (which the authors assumed didn't happen in the comments on neglecting backscattering)."

-> Concerning phonons and backscattering, we think that there is a confusion on this matter. When the system is in contact with a thermal bath, we do assume in sections 5.1 that the bath is indeed thermalizing both left and right movers at a same temperature. Hence we get an effective thermalization between left and right movers.

When we consider a quantum quench, once the external temperature is released, the conductor is no longer in contact with any thermal bath. In that case, we do not assume a local thermalization between the left and right movers (as opposed as between them and an external bath). In other words, we assume left and right movers to evolve ballistically and hence not to thermalize with each other. In absence of such an hypothesis the left and right moving heat waves would be damped due to the thermalization.

---

## Round 2 · Author Response

Response to report of Referees

We thank the Referees for providing us with detailed reports. Below, we address the concerns raised and describe the changes introduced in the manuscript (all modifications, except for the additions in the bibliography, are marked in {\color{red}red text}). For the sake of convenience, we address the remarks of the Referee in quoting the full texts of the reports. For the same reason, we took a liberty to divide the report in parts, and answer them part by part.

REFEREE 1
Report:"The manuscript discusses corrections to Luttinger relations due to quantum anomalies. It is clearly written, and the computations are easy to follow."

Our response"We thank the Referee for the positive evaluation of our work."

Report:"In my opinion the only aspect of the manuscript that needs to be expanded is the discussion that the particles transport ballistically, and the two species don’t thermalize. This assumption for thermal systems seems to be very strong and in general valid for temperatures close to zero. Therefore, I would expect that it breaks down quite quickly as the temperature rises. However, I have not seen any discussion of the applicability of the formulas presented in the paper."

Our response "We thank the referee for pointing out that a discussion of the validity of our results was needed. Indeed, this is crucial. We have modified the section 6 of our manuscript, which now starts by a description
of the condition of application of our approach. We indeed require that heat transport is ballistic and that the temperature of carriers varies along their path, which means that some energy relaxation occur.

Let us comment on the first requirement: in strictly 1D, there is no diffusive regime. Transport is either ballistic, or if backscattering occurs it opens a gap at the chemical potential, corresponding to the Anderson localization (see e.g. {\it Quantum Physics in One Dimension}, Th. Giamarchi, OUP). Hence for strictly 1D systems, we want this backscattering to be negligible.
For quasi-1D systems with a number $N$ of modes, such as quantum wires of metals in a strong magnetic field,
the corresponding localization length scales like $\propto N l_e^{(b)}$ where $l_e^{(b)}$ is the {\bf elastic backscattering length}. In that situation, we only require that the backscattering potential is sufficiently weak such that $ N l_e^{(b)} \ll L$ where $L$ is the sample size. Note that in such a situation, the weak backscattering can be effectively accounted for as a redistribution of energy between left and right movers. We do not expect any qualitative change of our results for $l_e^{(b)} \ll L \ll N l_e^{(b)}$ although a quantitative description of this situation requires additional work beyond that of the present manuscript.

The second requirement is that the distribution of local observables is well approximated by a thermal Maxwell–Boltzmann distribution parametrized by a local temperature $T(x)$. Such a local distribution requires energy distribution within the ensemble of left or right moving excitations. This corresponds to a short {\bf inelastic forward scattering length} $l_e^{(f)} \ll L$. Our formalism describes the behavior of energy distribution and currents on scales intermediate between $l_e^{(f)}$ and $L$.

In conclusion: from these two conditions, we require $l_e^{(f)} \ll L \ll N l_e^{(b)}$. The first length $l_e^{(f)}$ depends on the $q\simeq 0$ Fourier components of the scattering potential, while the second length $l_e^{(b)}$ is set by its $q\simeq 2 k_F$ components. In most thermal conductors, including the "low density semi-metals", $k_F$ is a sufficiently large momentum to allow for a separation of length scales associated to both Fourier components. In these realistic situations, the backscattering originates from local impurities, while forward scattering originates from {\it e.g.} coupling to acoustic phonons, non-linearities of the spectrum, etc. Hence thermal transport in a clean conductor of mesoscopic size, {\it e.g.} $(10-100) \mu$m, would realistically correspond to the situation we describe.

Finally, a last assumption in part of our paper, including the black hole and the ballistic response theory section, is that there is a single local temperature $T(x)$ common to both left and right movers.
This assumption is driven by concerns of simplification and pedagogy. In general, we do not expect the left and right particles to equilibrate with each other on short lengthscales: this would require efficient inelastic backscattering. Indeed, close to a blackhole, only the outgoing energy current sets the asymptotic Hawking radiation. The equilibration with an ingoing energy current is by no mean required. Indeed we have been very careful to express our results as a sum of contributions from left and right movers. In the general case, all of our results of sections 3 and 4 still apply for the left and right movers, possibly with two independent temperature profiles . On the other hand, the case of the quantum quench of section 5 is perfectly fine: the local temperature is set externally, through a coupling to both left and right movers. So this situation does not require additional conditions.

MODIFICATION: We have added a discussion of the applicability of our result at the
beginning of the discussion in section 6."

Report:"Moreover, the validity of the corrections derived in this work requires that different species of fermions are kept at different temperatures. Again, no discussion of how close to realistic systems is this requirement, is included."

Our response:"We believe that the referee refers to the quantum quench dynamics described in section 5.2 of our manuscript as this is the situation we describe where left and right moving particles evolve at two different temperatures. As we described in answer to the previous point raised by the referee, it is rather natural to expect that backscattering is weak in most thermal conductors we have in mind for potential experimental measurement. Yet, the physics described in this section lies at the forefront of what is currently experimentally accessible. In a simple experimental protocol, where two pieces of quantum conductor are maintained at two temperatures $T_L$ and $T_R$ and put in contact at $t=0$, measuring the heat waves would require fast time-resolved measurements of heat flux, able to resolve the heat waves. To our knowledge, such experimental probes are currently being developed but at not yet available. Among them, the so-called Pump-Probe microscopy techniques are very promising,
see for example \href{https://doi.org/10.1016/j.chemphys.2015.07.006}{j.chemphys.2015.07.006} and \href{https://journals.aps.org/prb/abstract/10.1103/PhysRevB.102.184307}{Phys. Rev. B 102, 184307 (2020)}.
They consist in heating locally a material with a laser pulse, and probing the dynamics of the excitations by measuring the diffusion of a probe signal. For a weak electron-phonon couplings, the dynamics of the electronic-excitation will be described by a quench procedure aking to that of our manuscript."

MODIFICATION: We have added a discussion of the applicability of our result at the
beginning of section 6.

Report:"I do not expect these relations to hold near black holes."

Our Response:"Within the standard description, the Hawking radiation close to a black hole originates from the massless vacuum fluctuations. These do not scatter on each other, and are believed to be at local thermal equilibrium.
However, we stress that the discussion of the black hole physics in our manuscript serves as a motivation to explore analogous physics in condensed matter. "

MODIFICATION: in the discussion, we stress explicitly that the condition of applicability of our results focused exclusively on the condensed matter situations

REFEREE 2

Report:"This paper discusses nonlinear effects due to gravitational anomalies in 1+1d CFTs in a curved space, and the relationship with Luttinger's perspective on the relation between temperature gradients and gravity. My honest sense is that this paper is in some sense a commentary on well-established ideas about thermal CFTs in a curved background, and remarking on the breakdown of the calculation from Luttinger's linear response calculation? That could be valuable to the literature."

Report:"I think my main concern - or at least confusion - about this manuscript is the following. A notion of local temperature is probably only well-defined in curved spacetime with slowly varying curvature. Equilibrium is sharply defined: I can ask what is the profile of $<T^{\mu \nu}>$ given a classical gravitational background, and the anomalous energy-momentum equation subject to boundary conditions. Assuming no quantum gravity, I can exactly solve those equations without ambiguity. But then, it is far from clear to me whether particular experimental probes of temperature correspond to what the authors think they do? Some possibly semantic questions about whether some particular definition of temperature need to be corrected at the nonlinear level, might not be that interesting to me personally, because I do not know whether the authors' definition of temperature is the one probed by any given experimental system. As one example, the discussion around Eq. 56 and 57 to me seems very possibly compatible with this idea: I start with some sharply varying temperature profile in flat space and, depending on interpretation, may assign it different notions of "local temperature"?"

Our response:"We thank the referee for stressing that our initial discussion of equilibrium temperature versus an imposed external one was very confusing. Indeed, since the initial submission of our manuscript, we have critically reconsidered this notion of two local temperatures. And we have realized that the notion of "external temperature" used in the previous version of the paper
was erroneous. In any system, there can be at most a single local temperature. This has led us to realize that in a thermal generalized Gibbs measure, the prefactor $\beta(x)$ of the local energy density in the exponential weight does not set solely the inverse local temperature, but has to be complemented by anomaly corrections. We have now reversed the method: given a local equilibrium temperature, imposed by contact with external baths, what is the corresponding function $\beta(x)$ and associated generalized
Gibbs ensemble that sets the local physical observables. These results have important consequences for the physics of thermal extended Gibbs ensemble.

MODIFICATION: We have completely rewritten the section 5.1 and 5.3 of our manuscript, and we have modified the Figures 3, 4 and 5, and removed the figure of the phase diagram.

Report:"What an experiment could clearly (in principle) measure are the correlation functions of $<T^{\mu \nu}>$, which the authors calculate. But then, this calculation seems to be based on an assumption of how an external temperature profile would couple to the theory, and again, I am not sure how confident I am in that theory on arbitrarily short length scales. Maybe I am the one who does not understand."

Our response: "We thank the referee for pointing out that a discussion of whether our prediction can be tested experimentally, which correlated with a similar question of Referee 1. Indeed, these questions are crucial. We indeed require that heat transport is ballistic and that the temperature of carriers vary along their path, which means that some energy relaxation occur.

Let us comment on the first requirement : in strictly 1D, there is no diffusive regime. Transport is either ballistic, or if backscattering occurs it opens a gap at the chemical potential, corresponding to the Anderson localization (see e.g. {\it Quantum Physics in One Dimension}, Th. Giamarchi, OUP). Hence for strictly 1D systems, we want this backscattering to be negligible. For quasi-1D systems with a number $N$ of modes, such as quantum wires of metals in a strong magnetic field,the corresponding localization length scales like $\propto N l_e^{(b)}$ where $l_e^{(b)}$ is the {\bf elastic backscattering length}. In that situation, we only require that
the backscattering potential is sufficiently weak such that $ N l_e^{(b)} \ll L$ where $L$ is the sample size. Note that in such a situation, the weak backscattering can be effectively accounted for as a redistribution of energy between left and right movers. We do not expect any qualitative change of our results for $l_e^{(b)} \ll L \ll N l_e^{(b)}$ although a quantitative description of this situation require additional work beyond that of the present manuscript.

The second requirement is that the distribution of local observables is well approximated by a thermal Maxwell–Boltzmann distribution parametrized by a local temperature $T(x)$. Such a local distribution requires energy distribution within the ensemble of left or right moving excitations. This corresponds to a short {\bf inelastic forward scattering length} $l_e^{(f)} \ll L$. Our formalism describes the behavior of energy distribution and currents on scales intermediate between $l_e^{(f)}$ and $L$.

In conclusion: from these two conditions, we require $l_e^{(f)} \ll L \ll N l_e^{(b)}$. The first length $l_e^{(f)}$ depends on the $q\simeq 0$ Fourier components of the scattering potential, while the second length $l_e^{(b)}$ is set by its $q\simeq 2 k_F$ components. In most thermal conductors, including the "low density semi-metals", $k_F$ is a sufficiently large momentum to allow for a separation of length scales associated to both Fourier components. In these realistic situations, the backscattering originates from local impurities, while forward scattering originates from {\it e.g.} coupling to acoustic phonons, non-linearities of the spectrum, etc. Hence thermal transport in a clean conductor of mesoscopic size, {\it e.g.} $(10-100) \mu$m, would realistically correspond to the situation we describe.

Finally, a last assumption in part of our paper, including the black hole and the ballistic response theory section, is that there is a single local temperature $T(x)$ common to both left and right movers. This assumption is driven by concerns of simplification and pedagogy. In general, we do not expect the left and right particles to equilibrate with each other on short lengthscales: this would require efficient inelastic backscattering. Indeed, close to a blackhole, only the outgoing energy current sets the asymptotic Hawking radiation. The equilibration with an ingoing energy current is by no means required. Indeed we have been very careful to express our results as a sum of contributions from left and right movers. In the general case, all of our results of sections 3 and 4 still apply for the left and right movers, possibly with two independent temperature profiles. On the other hand, the case of the quantum quench of section 5 is perfectly consistent: the local temperature is set externally, through a coupling to both left and right movers. So this situation does not require additional conditions.

MODIFICATION: We have added a discussion of the applicability of our result at the beginning of the discussion in section 6.

Report:"I am confused around Eq. (53). Are the authors assuming that the temperature gradient is constant while the energy current is not? That seems to require a very special type of heat bath. It could just as easily be that the actual temperature varies in some nonlinear way, so that the energy current is better behaved. I do not see how this discussion can feasibly relate to most experimental systems, where if anything, it seems more plausible to argue some systems are approximately thermally isolated when electron-phonon coupling is weak. In this setting it is $J_\varepsilon$ that must be constant, while $T(x)$ could vary quite nonlinearly. If the system is not thermally isolated, then the thermodynamics of the phonon bath must also be accounted for?"

Our response:"Indeed, Eq.(53) describes the heat current induced by a temperature gradient beyond the linear response theory. This theory describes the first order of an expansion, which is valid when the thermal gradient is small. When larger temperature gradient are imposed, additional non-linear terms have to be incorporated in the response theory. We describe the situation where the temperature profile remains linear. When non-linearities of the temperature are considered, we have to resort to a full analogy with a gravitational potential, described in section 5. "

Report:"In Section 5.2, the authors have a discussion of dynamics after a quench. From my perspective this sounds like a very straightforward problem where gravity is not needed at all: there is a $T_L$ and $T_R$ associated to the left and right movers of the CFT, and the profiles of $T_L$ and $T_R$ simply propagate at velocity c. If this is all the authors are commenting on from the gravitational perspective, it would be helpful to understand the role that the anomaly is playing more carefully by adding some concrete equations showing the imprint of the anomaly after the spacetime becomes flat? For example, if I encode some initial inhomogeneity in energy density/energy current via coupling to background spacetime, and then quickly shut off this coupling and return to flat space, presumably once the coupling is shut off the anomalous terms disappear from the energy momentum equation. Then, is there any anomalous part to the energy-momentum tensor after the quench? I fail to see how it is possible, once the quench ends the spacetime would be flat? Perhaps again, the only way the anomaly is encoded is in the precise way the theory is coupled to the external heat bath, and it's far from clear to me that there is only one logical way to couple to this heat bath, especially if the proposed temperature varies on extremely short length scales."

Our response:"Indeed for time $t>0$, the spacetime is Flat and present no gravitational anomaly. The profiles, determine at time t=0 simply propagates to the left and the right at a velocity c. However, this quench procedure allows one to observe signatures of the gravitational anomalies present at time $t<0$ on the energy current $J_\varepsilon$. The energy current profiles propagating at time $t>0$ are indeed carrying a signature of the gravitational anomaly characterized by the blue area in Figure 4.b and 4.c ."

Report:"I think this commentary should be addressed or refuted in a revised manuscript, but I would probably encourage publication afterwards."

Our response:"We believe that we have answered thoroughly the concerns of the Referee, and we hope that he will be convinced that our modified manuscript is now suitable for publication."

We believe that our manuscript, in its revised form, is suitable for publication.

Sincerely Yours,

The Authors

---

## Round 2 · List of Changes

All the changes appear in red in the resubmission file.

In more details:

-In the abstract: "induced by repeated thermal quenches"->"induced by repeated thermal quenches"

-p4: "periodic sequence of temperature quenches"->"periodic sequence of metric quenches"

-p6, below equation 4: "Upon spatial differentiating we recover the Luttinger relation~\eqref{eq:Luttingerrel}
to lowest order in the gravitational factor $\phi$." ->"Upon spatial differentiating with respect to position we recover the Luttinger relation ~\eqref{eq:Luttingerrel} "

-p9 we changed the title of the paragraph: "Lorentz anomaly"->"Covariantly conserved tensor"

-p11 we changed the title of the paragraph: "Anomalous momentum-energy tensor breaking Lorentz invariance."->"Anomalous covariantly conserved momentum-energy tensor"

-p11 below equation 32: "where $T_0$ is the reference temperature introduced Eqs.~\eq{eq:Tolman} and \eq{eq:TTE}"->"where $T_0$ is the reference temperature introduced Eqs.~\eq{eq:Tolman} and \eq{eq:TTE}, and which is now chosen as $T_0 = T(x_0)$ at a point $x_0$ such that $\phi(x_0)= \partial_x^2 \phi(x_0) = 0 $."

-p13 in the caption of figure 2:"are represented as a function of the distance $x$ to the horizon $x_H$" ->"are represented as a function of the distance $x$ to the center of the black hole, in units of its horizon $x_H$"

p15: "Its asymptotic value satisfies $ \varepsilon = -p = (\hbar c / 96\pi) \partial_x^2 f(x_H) <0 $, reminiscent of the Casimir effect \cite{casimir1948attraction,milton2001casimir}. "->"with an asymptotic value $ \varepsilon = -p = (\hbar c / 96\pi) \partial_x^2 f(x_H) <0 $."

p18: "we focus on the situation of a quantum system "->"we focus on the situation of a finite size quantum system "

Section 5.1 was entirely rewritten and figure 3 was changed accordingly

Figure 4 p22 was modified to fit with the new figure 4 and to describe a temperature quench

To be coherent with the notation of section 5.1, section 5.3 and the caption of Figure 5 were reformulated

We added a few paragraph at the beginning of the Discussion section, to discuss the condition of application of our approach

Missprint were corrected in the Appendix pages 30 and 31 with equations (96) and (100)

---

## Editorial Decision

published